# Mapping *cis*- and *trans*-regulatory target genes of human-specific deletions

Tyler Fair[1,2,3], Bryan J. Pavlovic[1,3], Dani Swope[1,3], Octavio E. Castillo[4], Nathan K. Schaefer [1,3] & Alex A. Pollen [1,3] ✉

Deletion of functional sequence is predicted to represent a fundamental mechanism of molecular evolution. Comparative genetic studies of primates have identified thousands of human-specific deletions (hDels), and the *cis*-regulatory potential of short (≤31 base pairs) hDels has been assessed using reporter assays. However, how structural variant-sized (≥50 base pairs) hDels influence molecular and cellular processes in their native genomic contexts remains unexplored. Here, we design genome-scale libraries of single-guide RNAs targeting 7.2 megabases of sequence in 6358 hDels and present a systematic CRISPR interference (CRISPRi) screening approach to identify hDels that modify cellular proliferation in chimpanzee pluripotent stem cells. By intersecting hDels with chromatin state features and performing single-cell CRISPRi (Perturb-seq) to identify their *cis*- and *trans*-regulatory target genes, we discovered 20 hDels controlling gene expression. We highlight two hDels, hDel_2247 and hDel_585, with tissue-specific activity in the brain. Our findings reveal a molecular and cellular role for sequences lost in the human lineage and establish a framework for functionally interrogating human-specific genetic variants.

Millions of single-nucleotide and structural variants (SVs)—deletions, duplications, insertions, and inversions ≥50 base pairs (bp) in length—have accumulated in the human lineage since divergence from non-human primates[1–4]. Contained within this genetic variation are alterations to functional sequences that distinguish humans from nonhuman primates[5,6]. However, the overwhelming majority of variants are predicted to be selectively neutral[7]. Among polymorphic variants, deletions are enriched for driving splicing and expression quantitative trait loci[8] and noncoding deletions predicted to be deleterious exhibit levels of purifying selection comparable to loss-of-function coding alleles[9]. Noncoding deletions removing *cis*-regulatory elements may also underlie instances of adaptive evolution[10,11]. Recent studies have identified noncoding deletions in the human genome and assessed their capacity for driving reporter gene expression[1,6,12]. We reasoned that inactivating human-specific deletions (hDels) using tiling CRISPRi-based genetic screens in chimpanzee cells would enable systematic genome-scale interrogation of the effect of this class of genomic alterations on cellular proliferation and gene expression.

## Results

We focused on 7278 SV-sized (≥50 bp) hDels previously identified through comparison of long-read great ape genomes[1] ("Methods"). hDels span 12.7 megabases (Mb) of the chimpanzee reference genome (panTro6), are a median of 626 bp (range 50–262,923 bp), and primarily intersect noncoding regions (52.4% of hDel base pairs are intronic, 47.4% are intergenic, and 0.2% are exonic). Compared with matched genomic sequences[13], hDels are enriched for repeat elements ($p < 10^{-3}$, 63.2% of hDel base pairs are repetitive) and intergenic regions ($p < 10^{-3}$) and depleted from introns ($p = 7 \times 10^{-3}$) and exons ($p < 10^{-3}$). While hDels are depleted for overlap with conserved sequences, 2177

[1]Eli and Edythe Broad Center of Regeneration Medicine and Stem Cell Research, University of California, San Francisco, San Francisco, CA, USA. [2]Biomedical Sciences Graduate Program, University of California, San Francisco, San Francisco, CA, USA. [3]Department of Neurology, University of California, San Francisco, San Francisco, CA, USA. [4]Quantitative Biosciences Institute, University of California, San Francisco, San Francisco, CA, USA. ✉e-mail: alex.pollen@ucsf.edu

hDels remove sequences under purifying selection at levels comparable to exonic sequence (116,828 bp, depletion $p < 10^{-3}$). To characterize the epigenetic state of chromatin at hDels present in the chimpanzee reference genome, we performed Omni ATAC-seq[14] (Supplementary Fig. 1) in chimpanzee induced pluripotent stem (iPS) cells from four individuals (C3624K, C3651, C8861, Pt5-C) and profiled H3K4me1[ab8895], H3K4me3[ab8580], H3K27ac[ab4729], and H3K27me3[9733S] histone modifications using CUT&Tag[15] (Supplementary Fig. 2). Although hDels are depleted from Tn5-accessible and H3K4me1-, H3K4me3-, H3K27ac-, and H3K27me3-modified chromatin ($p < 10^{-3}$), we identified 290 hDels intersecting at least one of these epigenetic features, revealing sequences lost in the human lineage harboring candidate cis-regulatory elements.

To evaluate the functions of hDels in their native genomic contexts using a CRISPRi-based genetic screening approach, we first introduced dCas9-KRAB into the *CLYBL* safe harbor locus[16] in iPS cells from two male chimpanzees (Supplementary Fig. 3a, C3624K, Pt5-C). We then designed a library of sgRNAs tiling across all hDels (hDel-v1) and separately, a library targeting hDels intersecting epigenetic features associated with cis-regulatory elements (hDel-v4).

To probe the effect of hDels as a class of human-specific SVs on a quantitative cellular phenotype, we designed a library of 170,904 sgRNAs tiling across 7.2 Mb of sequence within 6358 hDels (Supplementary Fig. 3b, hDel-v1). We considered hDels independent of evolutionary conservation or epigenetic state, as these features may not be predictive of all classes of cis-regulatory elements[17]. To tile across all uniquely targetable hDels, sgRNAs were assigned to 50-bp genomic windows and the sgRNA with the highest predicted activity[18] per window was selected for inclusion in hDel-v1 (Fig. 1a, median 52 bp between sgRNAs, median 14 sgRNAs per hDel). We reasoned that iPS cells would be a useful model for studying hDels because of their transcriptionally permissive chromatin structure[19] and sensitivity to proliferation-modifying perturbations. We transduced chimpanzee CRISPRi iPS cells (C3624K, Pt5-C) with the lentiviral hDel-v1 sgRNA library, selected for sgRNA-expressing cells with puromycin, cultured cells for 10 days, and quantified sgRNA enrichment and depletion by high-throughput sequencing (Fig. 1a–c).

Analysis of technical and biological replicates revealed that sgRNAs were highly correlated between replicates of the same cell line (Fig. 1d and Supplementary Fig. 3, Pearson's $r = 0.78$ to $0.88$) and

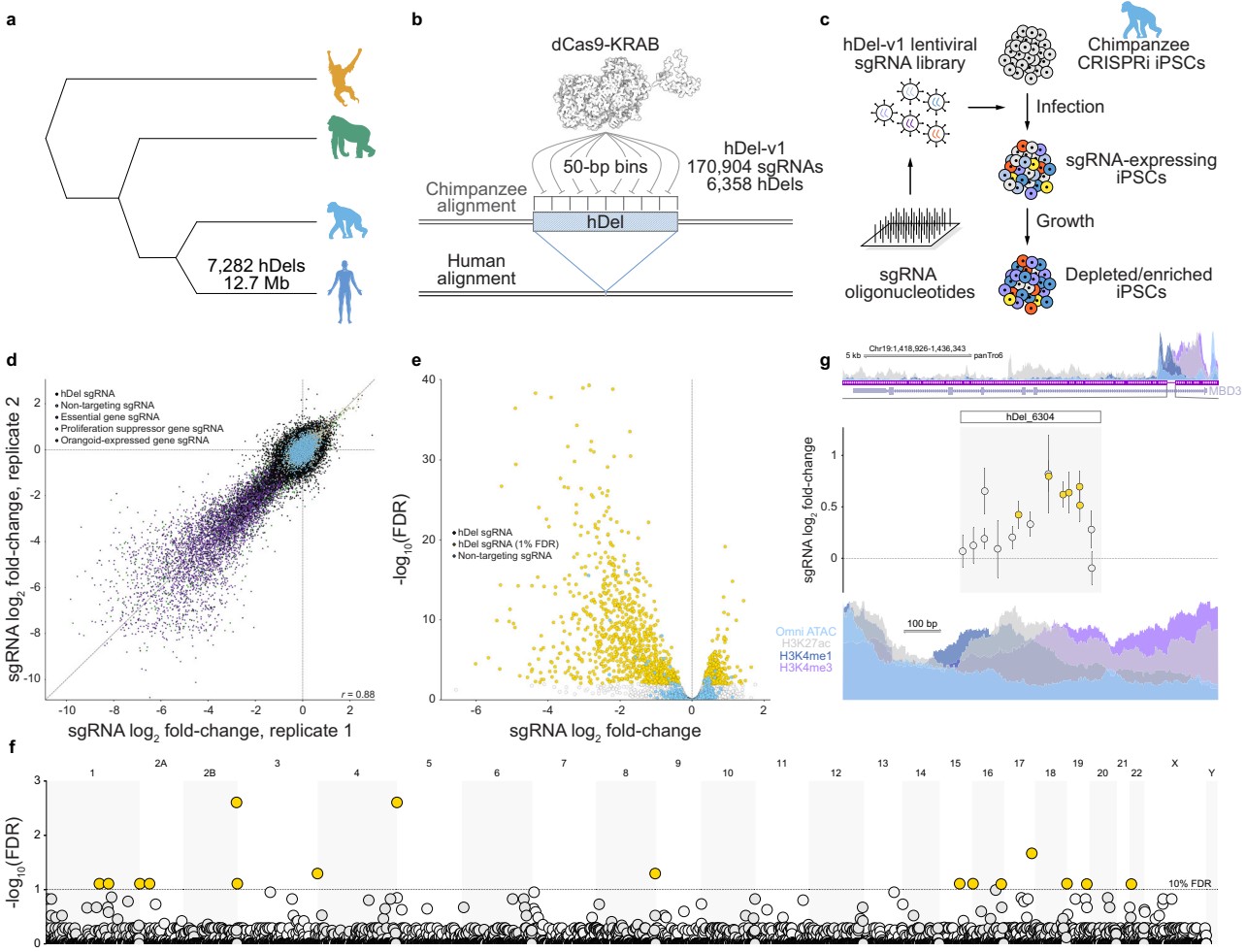

**Fig. 1 | Genome-scale tiling CRISPRi-based genetic screens identify hDels modifying cellular proliferation. a** Great ape cladogram. The number of deletions assigned to the human lineage[1] and the number of base pairs removed are labeled. **b** CRISPRi-based tiling of hDels (hDel-v1). hDel-v1 sgRNAs were selected from 50-bp genomic windows. **c** hDel-v1 screening approach in chimpanzee iPS cells. **d** Scatterplot of sgRNA log₂ fold-change for hDel-v1 technical replicates in C3624K. **e** Volcano plot of hDel-targeting and non-targeting sgRNA log₂ fold-change and DESeq2 Benjamini−Hochberg-adjusted *p*-value. **f** Manhattan plot of hDel position in the chimpanzee reference genome (panTro6) and α-RRA Benjamini−Hochberg-adjusted *p*-value for 500-bp hDel genomic windows (gold, FDR < 0.1). **g** hDel_6304-targeting sgRNA log₂ fold-change (gold, FDR < 0.05) and *MBD3* Omni ATAC-seq, H3K4me1, H3K4me3, and H3K27ac in C3624K. Data are the mean of two technical replicates ± standard error.

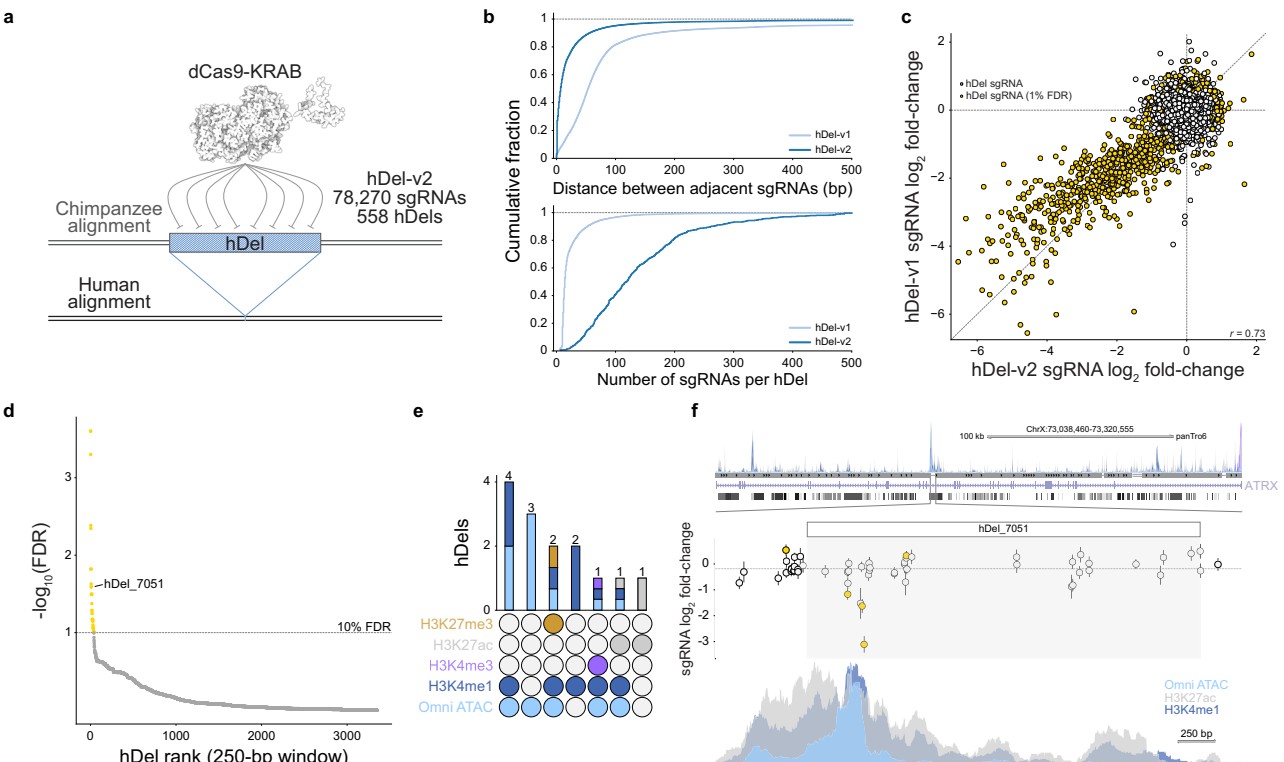

**Fig. 2 | High-density tiling CRISPRi screen refines the boundaries of functional sequences within proliferation-modifying hDels. a** High-density CRISPRi-based tiling of hDels (hDel-v2). **b** Distribution of the distance between adjacent hDel-targeting sgRNAs (top) and the number of sgRNAs per hDel (bottom) for hDel-v1 and hDel-v2. **c** Scatterplot of sgRNA $\log_2$ fold-change for hDel-targeting sgRNAs screened in hDel-v1 and hDel-v2 ($n$ = 18,148 sgRNAs). **d** Identification of proliferation-modifying hDels. 250-bp hDel genomic windows are ranked by α-RRA Benjamini–Hochberg-adjusted $p$-value (gold, FDR < 0.1). **e** Upset plot of Omni-ATAC seq, H3K4me1, H3K4me3, H3K27ac, and H3K27me3 intersecting hDels (FDR < 0.1) in C3624K. **f** hDel_7051-targeting sgRNA $\log_2$ fold-change (gold, FDR < 0.05) and *ATRX* Omni-ATAC seq, H3K4me1, H3K4me3, and H3K27ac in C3624K. Data are the mean of two technical replicates ± standard error.

between cell lines ($r$ = 0.69). As expected, sgRNAs targeting the promoters of essential or proliferation-suppressor genes were depleted or enriched, respectively (Fig. 1d). Using DESeq2[20] to model sgRNA counts from hDel-v1 and compute sgRNA false discovery rates (FDRs), we identified 1851 hDel-targeting sgRNAs modifying cellular proliferation (FDR < 0.01, Fig. 1e). Because sgRNAs exhibit variable CRISPRi activity depending on their genomic position, we assigned hDel-targeting sgRNAs to overlapping 500-bp genomic windows (250-bp step size) and computed hDel FDRs using alpha-robust rank aggregation (α-RRA). Genome-wide, we identified 16 hDels modifying cellular proliferation (FDR < 0.1, Fig. 1f), including hDel_6304, a 382 bp deletion located within the first intron of the nucleosome remodeling and deacetylase (NuRD) complex subunit MBD3, which increases cellular proliferation upon CRISPRi. (Fig. 1g).

To fine-map functional sequence within proliferation-modifying hDels identified using hDel-v1, we designed a second tiling library (hDel-v2) with reduced spacing between proximal sgRNAs. Because reduced spacing supports the discovery of hDels that may not have been detected using hDel-v1, we included hundreds of hDels with a single proliferation-modifying sgRNA. We omitted a genomic window strategy for sgRNA selection to maximize tiling density, resulting in a library of 78,270 sgRNAs targeting 558 hDels (Fig. 2a, b, median 7 bp between sgRNAs, median 119 sgRNAs per hDel). As with hDel-v1, we transduced chimpanzee CRISPRi iPS cells (C3624K) with the lentiviral hDel-v2 sgRNA library, selected and cultured sgRNA-expressing cells, and quantified sgRNA enrichment and depletion by high-throughput sequencing.

hDel-targeting sgRNAs screened in both hDel-v1 and hDel-v2 were highly correlated ($r$ = 0.73, Fig. 2c), as were technical replicates (Supplementary Fig. 4, $r$ = 0.82). We discovered 38 hDels modifying cellular proliferation (FDR < 0.1, Fig. 2d), including 14 hDels intersecting Omni ATAC-seq, H3K4me1, H3K4me3, H3K27ac, or H3K27me3, epigenetic features associated with *cis*-regulatory elements (Fig. 2e). For example, hDel_7051, a 2,602 bp deletion of a long interspersed nuclear element element within an intronic region of the SWI/SNF family chromatin remodeler *ATRX*, intersects Omni ATAC-seq, H3K4me1, and H3K27ac, and reduces cellular proliferation upon CRISPRi (Fig. 2f). Together, hDel-v1 and hDel-v2 identify cellular phenotypes for select hDels and indicate that despite the predicted phenotypic importance of SV-sized noncoding deletions[9], hDels as a class of human-specific SVs are largely dispensable for iPS cell proliferation.

We next sought to map the *cis*- and *trans*-regulatory targets of proliferation-modifying hDels using single-cell CRISPRi. To facilitate hDel-gene mapping, we designed a compact library of sgRNAs (hDel-v3) targeting hDels identified using hDel-v2 (122 sgRNAs targeting 19 hDels, FDR < 0.05). As positive and negative controls, we included transcription start site (TSS)-targeting sgRNAs, putative *cis*-regulatory element-targeting sgRNAs[21], and non-targeting sgRNAs (Supplementary Fig. 5a). We transduced chimpanzee CRISPRi iPS cells (C3624K) with the lentiviral hDel-v3 sgRNA library, selected and cultured sgRNA-expressing cells for 7 days, and performed single-cell RNA sequencing (Fig. 3a, Direct-capture Perturb-seq). After filtering, we recovered 16,810 sgRNA-expressing cells (Supplementary Fig. 5b–d, median 15,366 UMIs per cell, median 151 cells per sgRNA).

To identify the transcriptional consequences of hDel-v3 sgRNAs, we summed gene expression counts across cells containing each sgRNA and negative-control sgRNAs and performed a likelihood-ratio test using DESeq2. TSS-targeting sgRNAs mediated efficient target gene repression (Fig. 3b, 52.9 to 93.4%, median 85.0%, FDR < 0.1), demonstrating the performance of CRISPRi and the accuracy of

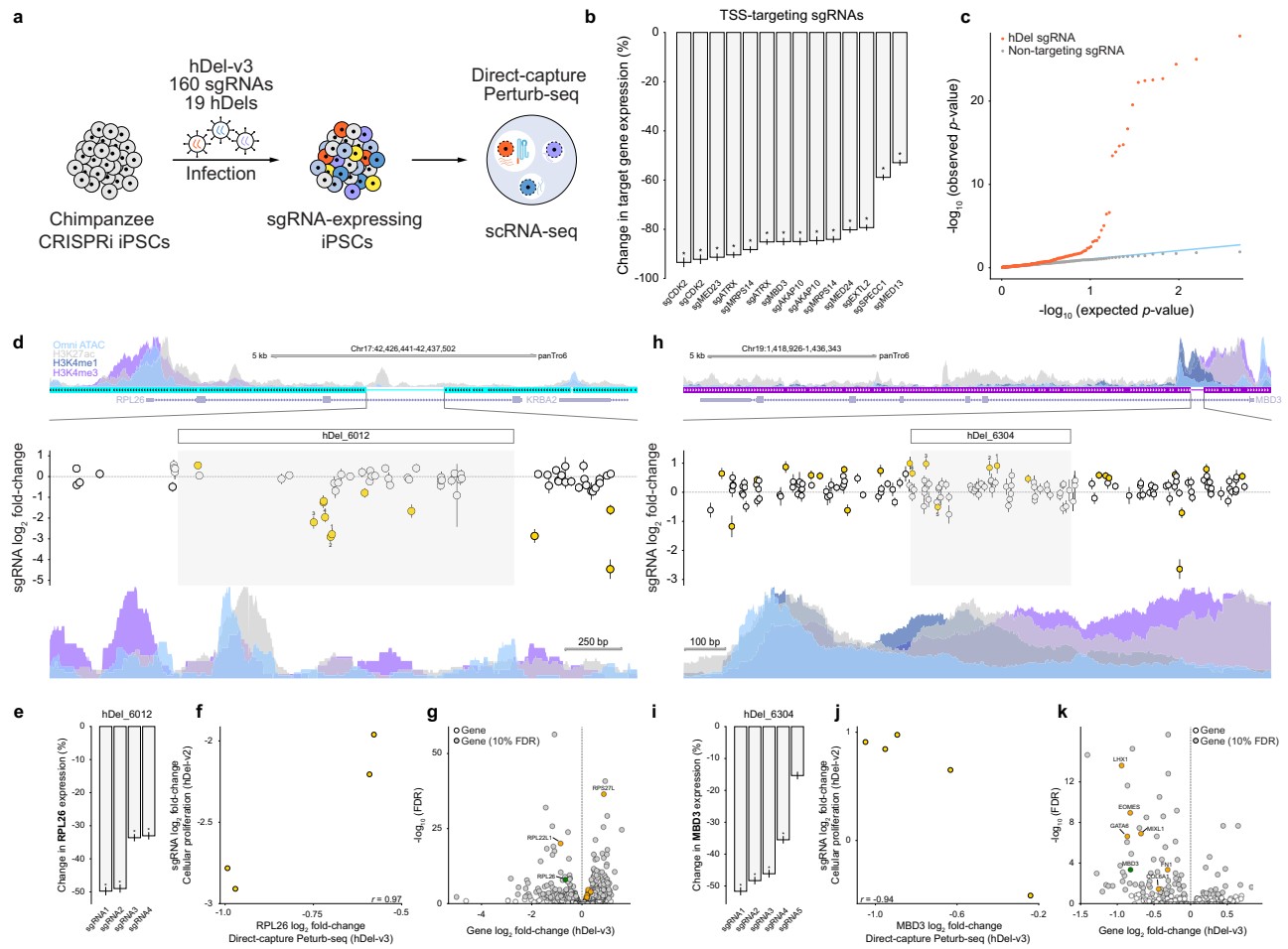

**Fig. 3 | Mapping *cis*- and *trans*-regulatory target genes of proliferation-modifying hDels. a** Single-cell CRISPRi screening approach (hDel-v3). **b** Differential gene expression for cells ("pseudobulk") harboring the indicated transcription start site-targeting sgRNA (*FDR < 0.1). Data are the mean of five technical replicates (GEM wells) ± standard error. **c** Distributions of observed and expected (uniform) *p*-values for *cis* differential expression for hDel-targeting sgRNA-gene pairs (orange) and non-targeting sgRNA-gene pairs (gray, down-sampled). Blue line: observed *p*-value = expected *p*-value. Two-sided *p*-values were computed from the normal cumulative distribution function using a gene-specific null distribution and adjusted for multiple comparisons using the Benjamini–Hochberg procedure. **d** hDel_6012-targeting sgRNA log₂ fold-change (hDel-v2; gold, FDR < 0.05) and *RPL26* Omni-ATAC seq, H3K4me1, H3K4me3, and H3K27ac in C3624K. Data are the mean of two technical replicates ± standard error. **e** Differential *RPL26* expression for cells harboring the indicated hDel_6012-targeting sgRNA (*FDR < 0.1). Data are the mean of five technical replicates (GEM

wells) ± standard error. **f** Scatterplot of hDel_6012-targeting sgRNA log₂ fold-change (cellular proliferation, hDel-v2) and *RPL26* log₂ fold-change (gene expression, hDel-v3). **g** *trans* differential expression for cells harboring hDel_6012-targeting sgRNAs (gene log₂ fold-change and Benjamini–Hochberg-adjusted *p*-value). Green: *cis* target gene; orange: ribosomal genes (FDR < 0.1). **h** hDel_6304-targeting sgRNA log₂ fold-change (hDel-v2; gold, FDR < 0.05) and *MBD3* Omni-ATAC seq, H3K4me1, H3K4me3, and H3K27ac in C3624K. Data are the mean of two technical replicates ± standard error. **i** Differential *MBD3* expression for cells harboring the indicated hDel_6304-targeting sgRNA (*FDR < 0.1). Data are the mean of five technical replicates (GEM wells) ± standard error. **j** Scatterplot of hDel_6304-targeting sgRNA log₂ fold-change (cellular proliferation, hDel-v2) and *MBD3* log₂ fold-change (gene expression, hDel-v3). **k** *trans* differential expression for cells harboring hDel_6304-targeting sgRNAs (gene log₂ fold-change and Benjamini–Hochberg-adjusted *p*-value). Green: *cis* target gene; orange: meso-endoderm (FDR < 0.1).

sgRNA-cell assignments. As expected, we observed an enrichment of significant sgRNA-gene pairs for hDel-targeting, but not non-targeting, sgRNAs (Fig. 3c). We identified 4 hDel-gene pairs within 100 kb (FDR < 0.1), including hDel_6012-*RPL26*, hDel_349-*MRPS14*, and hDel_6304-*MBD3*.

Targeting of hDel_6012 (Fig. 3d) reduced the expression of the 60S ribosomal protein L26 *RPL26* (Fig. 3e, 33.0 to 49.8%, median 41.3%, FDR < 0.1), and we observed a highly correlated relationship between the effect of hDel_6012-targeting sgRNAs on cellular proliferation and *RPL26* expression (Fig. 3f, *r* = 0.97). hDel_6012 does not intersect epigenetic features associated with *cis*-regulatory elements (Fig. 3d), highlighting the value of evaluating hDels independent of chromatin profiling. Genome-wide, we identified 636 differentially expressed genes upon hDel_6012 CRISPRi, including 35 genes encoding S and L ribosomal proteins, 32 of which (91.4%) were up-regulated (Fig. 3g,

FDR < 0.1). As expected, sgRNAs targeting hDel_6012 elicited correlated genome-wide transcriptional responses (Supplementary Fig. 6a, *r* = 0.57 to 0.68, FDR < 0.1).

sgRNAs targeting hDel_349 (Supplementary Fig. 6b) reduced the expression of the mitochondrial ribosomal 28S subunit *MRPS14* (Supplementary Fig. 6c, 60.3 to 88.3%, median 74.7%, FDR < 0.1), and we observed a highly correlated relationship between the effect of hDel_349-targeting sgRNAs on cellular proliferation and *MRPS14* expression (Supplementary Fig. 6d, *r* = 0.73). Genome-wide, we identified 11 differentially expressed genes upon hDel_349 CRISPRi, all of which correspond to mitochondrial or nuclear-mitochondrial transcripts (Supplementary Fig. 6e, FDR < 0.1), indicating that mitochondrial dysfunction underlies reduced proliferation. While we cannot exclude the possibility that hDel_349-targeting sgRNAs interfere with transcription at the *MRPS14* promoter, the sgRNAs reducing *MRPS14*

expression are located −302 to −658 bp relative to the TSS, outside of the optimal range of CRISPRi[22], and sgRNAs targeting *MRPS14* in human K562 cells[22] do not exhibit proliferation-modifying effects at similar distances (Supplementary Fig. 6f), suggesting that the observed effects are specific to hDel_349. *MRPS14* variants in humans are associated with muscle hypotonia, cognitive delay, and midface retrusion[23], providing evidence that loss of a *MRPS14 cis*-regulatory element may alter the development of multiple tissues.

Targeting of hDel_6304 (Fig. 3h) reduced the expression of the NuRD complex subunit *MBD3* (Fig. 3i, 35.5 to 51.7%, median 47.2%, FDR < 0.1). We observed an inverse relationship between the effect of hDel_6304-targeting sgRNAs on cellular proliferation and *MBD3* expression (Fig. 3j, *r* = −0.94), consistent with depletion of MBD3–NuRD inhibiting the differentiation of highly proliferative pluripotent cells[24]. Genome-wide, we identified 83 differentially expressed genes upon hDel_6304 CRISPRi, including transcription factors controlling meso-endoderm differentiation (*EOMES*, *GATA6*, *LHX1*, *MIXL1*), all of which were down-regulated (Fig. 3k, FDR < 0.1). hDel_6304 is also accessible in chimpanzee neural progenitor cells[25], suggesting that loss of hDel_6304 in the human lineage may contribute to increased neural stem and progenitor cell proliferation[26] by delaying terminal differentiation.

We next examined *RPL26*, *MRPS14*, and *MBD3* expression in human and chimpanzee cells. To quantify *cis*-regulatory divergence, we compared the expression of human and chimpanzee alleles in an identical *trans* environment. As expected, human alleles drove reduced *MRPS14* expression compared to chimpanzee alleles in human-chimpanzee allotetraploid iPS cells (*cis* contribution 13.1%, FDR < 0.05), explaining cross-species differences in gene expression (Supplementary Fig. 6h, 14.3%, FDR < 0.005)[27]. Similarly, *MBD3* expression is reduced in human, compared to chimpanzee, iPS cells (31.7%, FDR < 0.001)[28], and human alleles drove reduced *MBD3* expression (Supplementary Fig. 6i, *cis* contribution 34.7%, FDR < 0.15)[27]. Surprisingly, *RPL26* expression from human alleles is increased compared to chimpanzee alleles (Supplementary Fig. 6j, *cis* contribution 21.6%, FDR < 0.05)[27], indicating that additional *RPL26 cis*-regulatory alterations occurred in the human and chimpanzee lineages. Together, these findings suggest that hDel_349 and hDel_6304 remove *cis*-regulatory elements contributing to reduced *MRPS14* and *MBD3* expression in human cells, with *trans*-regulatory target genes controlling mitochondrial function and differentiation, respectively.

We also investigated potential sources of sgRNA off-target activity for proliferation-modifying hDels lacking *cis*-regulatory target genes. Because hDels are enriched for repeat elements, we focused on nucleotide homopolymers in hDel-targeting sgRNAs. Grouping hDel-v2 sgRNAs by the presence and position of $N_4$ homopolymers (AAAA, GGGG, CCCC) revealed that sgRNAs containing guanine ($G_4$) and cytosine ($C_4$) homopolymers exhibited pervasive off-target activity (Supplementary Fig. 7a, b). Off-target activity was dependent on the position of the homopolymer within the sgRNA, with the greatest toxicity observed for sgRNAs containing $G_4$ or $C_4$ near the center of the spacer sequence: sgRNAs containing $G_4$ at spacer position 9 were 5.0-fold more likely to have significant effects on cellular proliferation compared to all hDel-v2 sgRNAs (Supplementary Fig. 7c, d, 28.6%, FDR < 0.05), while sgRNAs containing $C_4$ at spacer position 11 were 2.4-fold more likely to have proliferation-modifying effects (Supplementary Fig. 7e, f, 13.7%, FDR < 0.05). Consequently, we excluded $G_4$/$C_4$-containing sgRNAs from hDel-v2 and hDel-v3 analysis. In hDel-v3, we did not observe correlated genome-wide transcriptional responses for homopolymer-containing sgRNAs, revealing no single transcriptional basis for proliferation-modifying effects. We also identified $G_4$-associated toxicity for sgRNAs tiling across *GATA1* and *MYC* in human K562 cells[29] (Supplementary Fig. 7g, h), providing evidence that sgRNA nucleotide homopolymers are an unappreciated source of off-target activity in CRISPRi-based genetic screens.

As hDels may control gene expression without directly modifying cellular proliferation, we focused on nonessential hDels intersecting epigenetic features associated with *cis*-regulatory elements. We designed a library of sgRNAs (hDel-v4) targeting hDels intersecting Omni ATAC-seq, H3K4me1, or H3K27ac (Fig. 4a, 888 sgRNAs targeting 163 hDels), including putative *cis*-regulatory element-targeting sgRNAs[21] and non-targeting sgRNAs as positive and negative controls, respectively (Supplementary Fig. 8a). To facilitate hDel-gene mapping using a larger library of sgRNAs, we transduced chimpanzee CRISPRi iPS cells (C3624K) with the lentiviral hDel-v4 sgRNA library at a high multiplicity of infection, selected and cultured sgRNA-expressing cells for 7 days, and performed single-cell RNA sequencing (Fig. 4a, Direct-capture Perturb-seq). After filtering, we recovered 18,571 sgRNA-expressing cells, detecting multiple sgRNAs per cell (Fig. 4b, Supplementary Fig. 8, median 7 sgRNAs per cell, median 495 UMIs per sgRNA per cell, median 26,989 UMIs per cell, median 195 cells per sgRNA).

To identify the transcriptional consequences of hDel-v4 sgRNAs, we summed gene expression counts across cells containing each sgRNA and all other cells and performed a likelihood-ratio test using DESeq2. As expected, sgRNAs targeting a putative inhibitor of DNA binding *ID1 cis*-regulatory element[21] reduced the expression of *ID1* (Supplementary Fig. 8d, 23.6 to 37.2%, FDR < 0.1). As with hDel-v3, we observed an enrichment of significant sgRNA-gene pairs for hDel-targeting, but not non-targeting, sgRNAs (Fig. 4c). We identified 16 hDel-gene pairs within 100 kb (Fig. 4d and Supplementary Fig. 8e, FDR < 0.1), including hDel_585-*HADHA*, hDel_1608-*C4orf48*, and hDel_2247-*PLPP1*.

hDel_585, a 207 bp intronic deletion located within the alpha subunit of the mitochondrial trifunctional protein *HADHA*, intersects Omni ATAC-seq (Fig. 4e). Although hDel_585 did not modify cellular proliferation (Fig. 4e), sgRNAs targeting hDel_585 reduced the expression of *HADHA* (Fig. 4f, 20.7 to 24.8%, FDR < 0.1). Because sequences within hDel_585 are not conserved to mice, we examined chromatin accessibility in additional nonhuman primates as a measure of *cis*-regulatory activity during development. Analysis of single-nucleus ATAC sequencing (snATAC-seq) of post-conception day 80 (PCD80) rhesus macaque prefrontal cortex (Supplementary Fig. 9, PFC) revealed that hDel_585 was accessible in radial glia, intermediate progenitor cells, and excitatory and inhibitory neurons (Fig. 4g). As deletions can increase *cis*-regulatory activity by removing transcriptional repressors or creating transcription factor binding sites[12], we performed Omni ATAC-seq of human, chimpanzee, and orangutan neuroepithelial cells to examine evolutionary differences in chromatin accessibility at the *HADHA* gene body. While hDel_585 was accessible in chimpanzee and orangutan neuroepithelial cells, we did not observe accessibility at the boundaries of hDel_585 in human neuroepithelial cells (Fig. 4h), consistent with the loss of *cis*-regulatory sequence. These results provide evidence that the loss of hDel_585 in the human lineage removed a *cis*-regulatory element active in the forebrain regulating *HADHA*.

hDel_1608, a 3.2 kb deletion located within the predicted lncRNA LOC104005955, intersects Omni ATAC-seq, H3K4me1, H3K4me3, and H3K27me3 (Fig. 4i). Targeting of hDel_1608 increased the expression of the Wolf-Hirschhorn Syndrome-associated[30] lumicrine factor[31] *C4orf48* (Fig. 4j, 16.2 to 19.8%, FDR < 0.1), which shares a bidirectional promoter with LOC104005955. This finding is consistent with the negative regulation of mRNA by a divergently transcribed lncRNA[32], although other mechanisms, including interference with repressive transcription factors could underlie increased expression following CRISPRi.

Several hDels linked to *cis*-regulatory target genes removed evolutionarily conserved sequences[6]. hDel_1273, linked to the GTP-specific beta subunit of succinyl-CoA synthetase *SUCLG2* (Supplementary Fig. 8g, 30.1%, FDR < 0.1), is partially conserved to platypus, and is

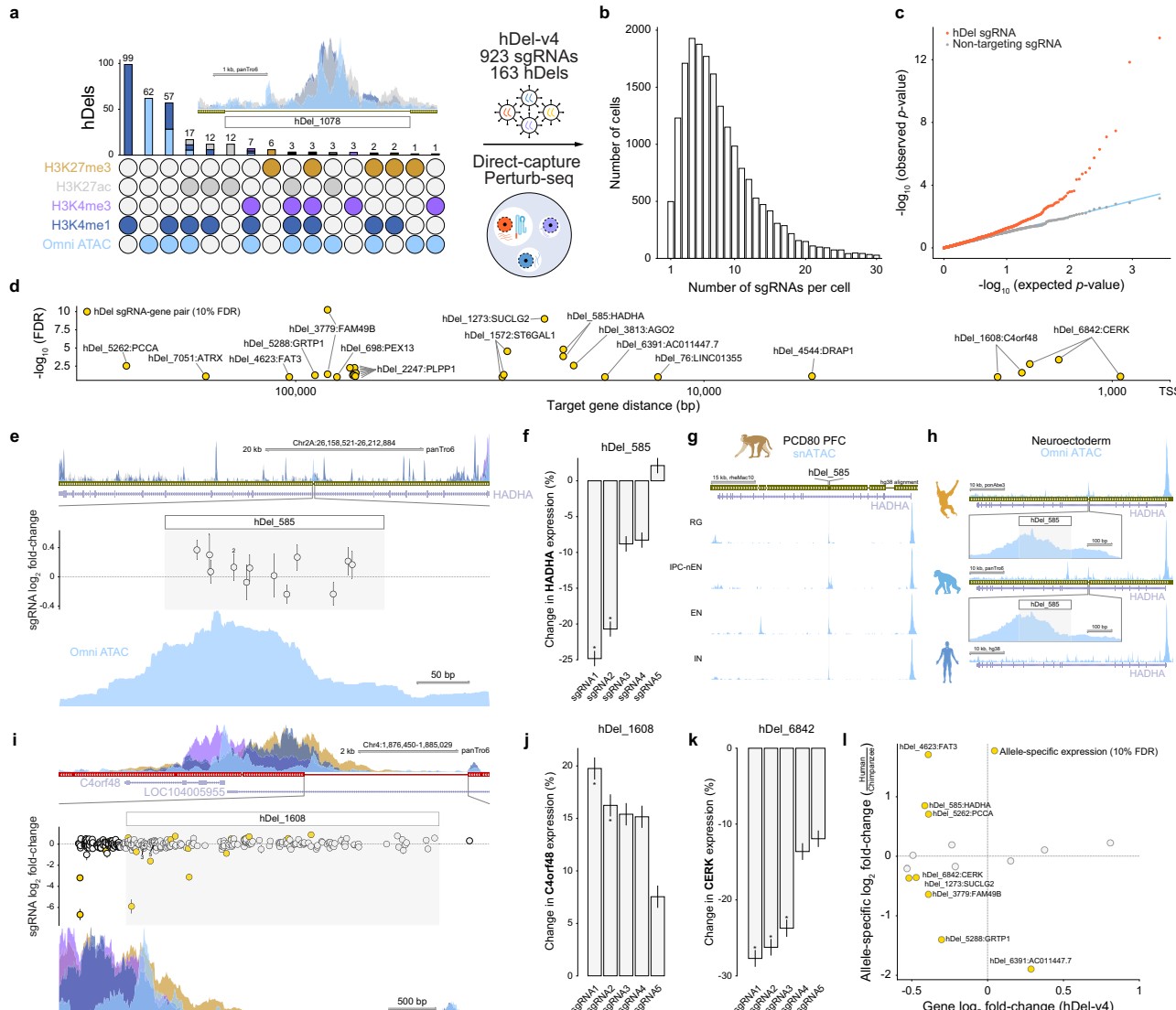

**Fig. 4 | Nonessential hDels harbor *cis*-regulatory elements. a** Upset plot of Omni ATAC-seq, H3K4me1, H3K4me3, H3K27ac, and H3K27me3 intersecting hDels and high multiplicity of infection single-cell CRISPRi screening approach (hDel-v4). Inset: hDel_1078. **b** Distribution of the number of sgRNAs per cell (hDel-v4). Shaded bar: median 7 sgRNAs per cell. **c** Distributions of observed and expected (uniform) *p*-values for *cis* differential expression for hDel-targeting sgRNA-gene pairs (orange) and non-targeting sgRNA-gene pairs (gray, downsampled). Blue line: observed *p*-value = expected *p*-value. Two-sided *p*-values were computed from the normal cumulative distribution function using a gene-specific null distribution and adjusted for multiple comparisons using the Benjamini–Hochberg procedure. **d** Distance between hDel-targeting sgRNA and TSS for corresponding *cis* target gene (FDR < 0.1). **e** hDel_585-targeting sgRNA log$_2$ fold-change (hDel-v1) and Omni ATAC-seq, H3K4me1, H3K4me3, and H3K27ac in C3624K. Data are the mean of two technical replicates ± standard error. **f** Differential *HADHA* expression for cells ("pseudobulk") harboring the indicated hDel_585-targeting sgRNA (*FDR < 0.1). Data are the mean of five technical replicates (GEM wells) ± standard error.

**g** snATAC-seq of PCD80 rhesus macaque prefrontal cortex at the *HADHA* gene body. RG radial glia, IPC-nEN intermediate progenitor cell-newborn excitatory neuron, EN excitatory neuron, IN inhibitory neuron. Shaded region: hDel_585 orthologous sequence. **h** Omni ATAC-seq in iPS cell-derived neuroepithelial cells from orangutan (top), chimpanzee (middle), and human (bottom) at the *HADHA* gene body. Shaded region: hDel_585 orthologous sequence. **i** hDel_1608-targeting sgRNA log$_2$ fold-change (hDel-v2) and Omni ATAC-seq, H3K4me1, H3K4me3, and H3K27me3 in C3624K. Data are the mean of two technical replicates ± standard error. **j** Differential *C4orf48* expression for cells harboring the indicated hDel_1608-targeting sgRNA (*FDR < 0.1). Data are the mean of five technical replicates (GEM wells) ± standard error. **k** Differential *CERK* expression for cells harboring the indicated hDel_6842-targeting sgRNA (*FDR < 0.1). Data are the mean of five technical replicates (GEM wells) ± standard error. **l** Scatterplot of allele-specific gene log$_2$ fold-change in human-chimpanzee allotetraploid iPS cells[27] and hDel-targeting sgRNA-gene log$_2$ fold-change (hDel-v4).

predicted to be active in the developing mouse heart, limb, midbrain[33]. Additionally, sgRNAs targeting hDel_3779, as well as complete genomic deletion of hDel_3779 using a pair of Cas9 ribonucleoproteins (RNPs), reduced the expression of the RAC1 effector *FAM49B* (Supplementary Fig. 8h and j, 14.2 to 23.6%, FDR < 0.1), a regulator of mitochondrial fission and cytoskeletal remodeling[34,35]. Finally, hDel_6842 intersects a mouse ENCODE proximal enhancer-like signature and is linked to the ceramide kinase *CERK* (Fig. 4k, 23.7 to 27.7%, median 26.2%, FDR < 0.1). CERK converts ceramide to ceramide

1-phosphate (C1P), a sphingolipid metabolite hydrolyzed by *PLPP1*[36], the *cis*-regulatory target of hDel_2247 (Fig. 4d), raising the possibility of epistasis between hDel_2247 and hDel_6842.

We also examined the relationship between hDels and human-chimpanzee *cis*-regulatory divergence. For *CERK*, *FAM49B*, *GRTP1*, and *SUCLG2*, human alleles drove reduced expression compared to chimpanzee alleles in allotetraploid iPS cells[27] (Fig. 4l, FDR < 0.1), consistent with the human-specific loss of *cis*-regulatory sequence. However, in certain cases, hDels removing *cis*-regulatory elements did not account

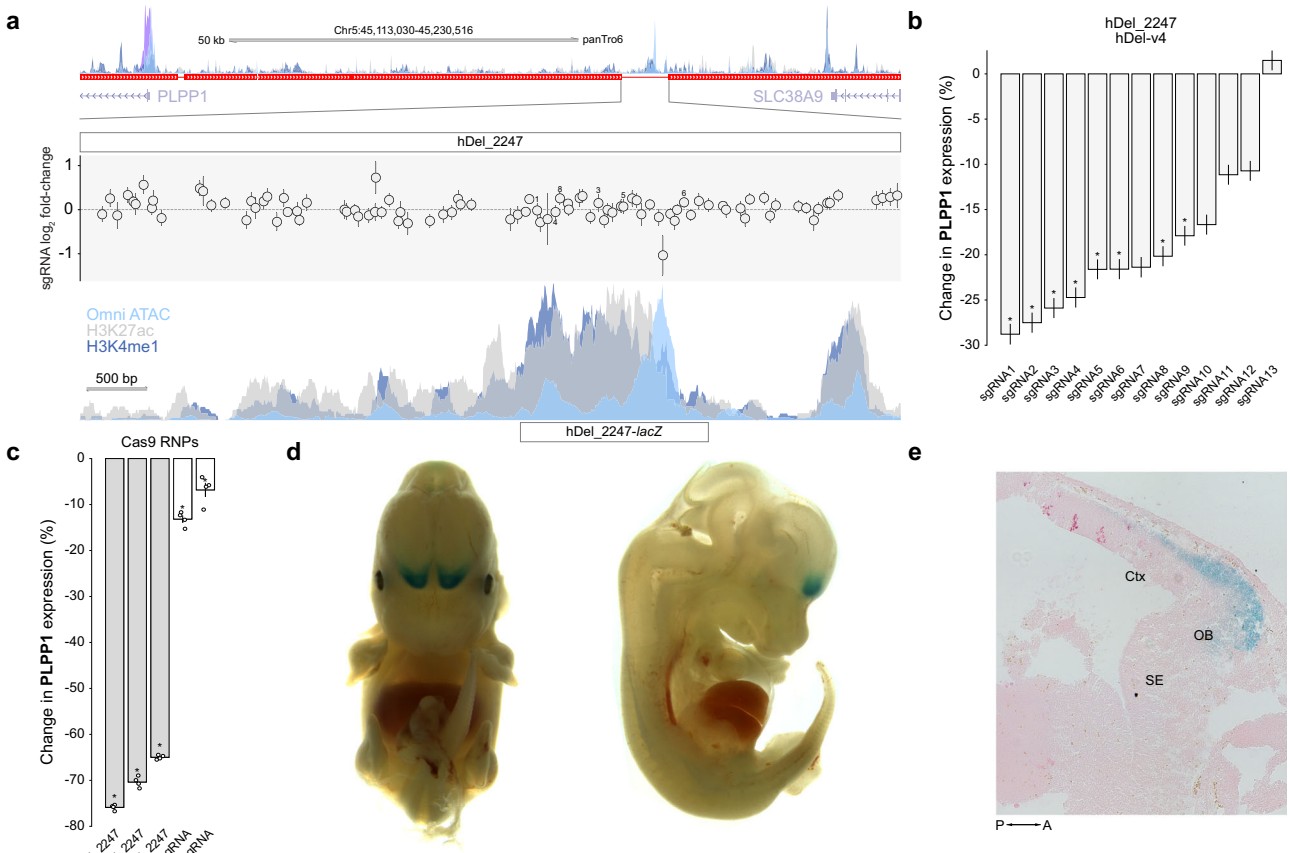

**Fig. 5 | hDel_2247 regulates *PLPP1* and drives *lacZ* expression in developing anterior cortex and olfactory bulb. a** hDel_2247-targeting sgRNA log$_2$ fold-change (hDel-v1) and Omni ATAC-seq, H3K4me1, H3K4me3, and H3K27ac in C3624K. Data are the mean of two technical replicates ± standard error. **b** Differential *PLPP1* expression for cells ("pseudobulk") harboring the indicated hDel_2247-targeting sgRNA (*FDR < 0.1). Data are the mean of five technical replicates (GEM wells) ± standard error. **c** Differential *PLPP1* expression (qRT-PCR) for

cells edited with hDel_2247-targeting Cas9 RNPs (*FDR < 0.05). Bars represent independent polyclonal populations of chimpanzee iPS cells (C8861). Data are the mean of four technical replicates ± standard error. **d** Representative E13.5 *hDel_2247::lacZ* mouse embryo stained for β-galactosidase (LacZ) activity (*n* = 5 embryos). **e** Sagittal section showing *lacZ* expression in olfactory bulb and anterior cortex. Ctx cortex, OB olfactory bulb, SE septum, P posterior, A anterior.

for cross-species *cis*-regulatory divergence. For example, *ATRX* was expressed at similar levels from human and chimpanzee alleles, and *HADHA* was expressed at higher levels from human alleles, suggesting that additional *cis*-regulatory alterations in the human and chimpanzee lineages attenuate or even reverse the *cis*-regulatory effects of select hDels, consistent with compensatory evolution within human-accelerated regions[37].

To further explore the functions of hDels with *cis*-regulatory activities, we focused on hDel_2247, a 6.8 kb intergenic deletion linked to the phospholipid phosphatase *PLPP1*. hDel_2247 is located between *PLPP1* (71 kb to TSS) and the lysosomal amino acid transporter[38] *SLC38A9* (107 kb to TSS) and intersects Omni ATAC-seq, H3K4me1, H3K27ac (Fig. 5a). Although hDel_2247 did not modify cellular proliferation (Fig. 5a), 8 sgRNAs targeting hDel_2247 reduced the expression of *PLPP1* (Fig. 5b, 17.9 to 28.8%, median 23.2%, FDR < 0.1). Moreover, complete genomic deletion of hDel_2247 using a pair of Cas9 RNPs resulted in a more pronounced reduction in *PLPP1* expression (65.0% to 75.9%; Fig. 5c and Supplementary Fig. 8j). To more closely examine the ancestral function of hDel_2247 in the human-chimpanzee last common ancestor, we synthesized a 1557 bp chimpanzee sequence intersecting Omni ATAC-seq, H3K4me1, and H3K27ac, and assessed its capacity to drive expression of a *lacZ* reporter gene in vivo. We found that the chimpanzee sequence drove consistent *lacZ* expression in the olfactory bulb and anterior neocortex (*n* = 4/5 embryos, Fig. 5d, e and Supplementary Fig. 10b, c),

two structures that have undergone morphological alterations in the human lineage. Reduced expression of *PLPP1* increases extracellular LPA in vivo[39], which may have cell-extrinsic effects, including on olfactory ensheathing cell migration[40] and neural stem and progenitor cell differentiation[41]. Together, these findings indicate that the loss of hDel_2247 in the human lineage removed a conserved *cis*-regulatory element regulating *PLPP1* with tissue-specific activity in the brain.

## Discussion

We established a systematic approach for evaluating how sequences lost in the human lineage modify cellular proliferation and gene expression by performing CRISPRi-based genetic screens in chimpanzee cells. Using libraries of sgRNAs tiling across 7.2 Mb of sequence within 6358 hDels, we assessed the effects of a class of human-specific SVs on a quantitative cellular phenotype. Although largely dispensable for proliferation, we identified hDels removing *cis*-regulatory elements controlling the expression of proliferation-modifying genes, including *MBD3*, *MRPS14*, and *RPL26*, and discovered the *cis*-regulatory target genes of 16 nonessential hDels intersecting Omni ATAC-seq, H3K4me1, and H3K27ac. Among nonessential hDels, hDel_2247 controls the expression of *PLPP1*, and its loss in the human lineage may alter phospholipid signaling in the developing olfactory bulb and prefrontal cortex. Several hDels linked to *cis*-regulatory target genes remove conserved sequences, and in certain cases, hDels account for

human-chimpanzee *cis*-regulatory divergence, underscoring the importance of deletions as a source of evolutionary innovation[5].

Our study provides a framework for applying CRISPRi-based forward genetic screens to characterize human-specific genetic variants in their native genomic contexts at scale[42]. Because it is difficult to predict which variants are functional, scalable approaches, such as genetic screens in cellular models, are required to systematically probe the millions of base pair alterations that have accumulated in the human lineage. In contrast to pooled reporter assays, which measure the capacity for variants to drive reporter gene expression in synthetic constructs[43], CRISPRi enables linkages to cellular phenotypes and genome-wide transcriptional responses. Recent CRISPRi-based studies have linked single-nucleotide polymorphisms identified by GWAS to target genes in *cis* and *trans* in disease-relevant cell types[44,45], but the functions of human-specific variants as a class of genomic alterations are not restricted to a single tissue or cell type. Pluripotent stem cells provide a useful model for screening variants with unknown tissue-specificities, as tissue-specific *cis*-regulatory elements may also be active in undifferentiated cells due to their transcriptionally permissive chromatin structure[19]. Additional CRISPRi-based mapping in cell types associated with morphological evolution[46,47] will facilitate interrogation of *cis*-regulatory divergence in the human lineage.

The adoption of CRISPRi-based screens for probing human-specific variants is subject to several design considerations. For modeling certain developmental differences, such as the expansion of neural stem and progenitor cell populations during human neurogenesis[26], proliferation represents a scalable and quantitative cellular phenotype, but does not directly reveal *cis*-regulatory element-gene linkages. Alternatively, single-cell CRISPRi enables high-dimensional molecular phenotyping[48], but may require variant selection using genetic, epigenetic, or transcriptomic features. Independent of phenotyping strategy, human-specific SVs frequently contain repeat-rich sequences[1], necessitating careful consideration of mismatched off-target sites[49] and $G_4/C_4$ nucleotide homopolymers during the design of sgRNA libraries. Among CRISPR-Cas modalities, CRISPRi facilitates fine-mapping functional sequences within SVs, while nuclease-active Cas9 enables the reconstruction of derived and ancestral alleles using sgRNA pairs, but at efficiencies that may be incompatible with pooled screening[50]. For *cis*-regulatory elements with repressive effects on transcription, such as silencers, CRISPR activation-based approaches may be useful for target gene identification.

In summary, the CRISPRi-based characterization of hDels presented here illuminates the loss of *cis*-regulatory elements in the human lineage and provides an approach, complemented by base and prime editors[51,52], for assigning molecular and cellular functions to all classes of human-specific genetic variants.

## Methods

### Sequencing
All Illumina sequencing was performed at the UCSF Center for Advanced Technology.

### Imaging
All imaging was performed at the UCSF Weill Imaging Core using a Leica THUNDER Imager.

### Cell lines and cell culture
Chimpanzee iPS cells from two healthy male donors (C3624K and Pt5-C) were cultured in v3.1 medium (65 ng/ml FGF2-G3, 2 ng/ml TGFβ1, 0.5 ng/ml NRG1, 20 μg/ml insulin, 20 μg/ml transferrin, 20 ng/ml sodium selenite, 200 μg/ml ascorbic acid 2-phosphate, 2.5 mg/ml bovine serum albumin (BSA), 30 ng/ml heparin, 15 μM adenosine, guanosine, cytidine, uridine nucleosides, 6 μM thymidine nucleosides in Dulbecco's Modified Eagle's Medium (DMEM)/Ham's F-12 (Corning,

10-092-CM)) supplemented with 100 IU/ml penicillin, and 100 μg/ml streptomycin at 37 °C and 5% $CO_2$. At ≥80% confluency, cultures were dissociated with phosphate-buffered saline (PBS) supplemented with 0.5 mM ethylenediaminetetraacetic acid (EDTA), resuspended in v3.1 medium supplemented with 2 μM thiazovivin (MedChemExpress, HY-13257), and seeded on Matrigel-coated cell culture plates (Corning, 354230). Protocols were approved by the Human Gamete, Embryo, and Stem Cell Research Committee at UCSF.

### CRISPRi iPS cell line generation
To establish chimpanzee iPS cell lines expressing dCas9-KRAB (*ZNF10/KOX1*), $1 \times 10^6$ iPS cells were seeded at a density of 100,000 cells/cm² and transfected with 3 μg pC13N-dCas9-BFP-KRAB (Addgene, 127968), 0.375 μg pZT-C13-L1 (Addgene, 62196), and 0.375 μg pZT-C13-R1 (Addgene, 62197) using 10 μl Lipofectamine Stem (Invitrogen, STEM00001)[53]. After 7 days, BFP-positive iPS cells were isolated by single-cell fluorescence-activated cell sorting. Transgene integration at the *CLYBL* locus was verified by PCR. To assess CRISPRi-mediated transcriptional repression, iPS cells were transduced with non-targeting or *SEL1L*-targeting sgRNAs[54], selected with 1.5 μg/ml puromycin (Gibco, A1113803), and processed for quantitative reverse transcription PCR.

### Lentivirus production and titration
Lenti-X HEK293T cells (Takara Bio, 632180) were maintained in DMEM/Ham's F-12 supplemented with 10% fetal bovine serum, 100 IU/ml penicillin, and 100 μg/ml streptomycin at 37 °C and 5% $CO_2$. To generate lentivirus, 150 mm cell culture dishes were coated with 10 μg/ml poly-D-lysine (Sigma-Aldrich, P7405) and Lenti-X HEK293T cells were seeded at a density of 85,000 cells/cm². The following day, medium was replaced, and Lenti-X HEK293T cells were transfected with 23.1 μg hDel sgRNA library transfer plasmid, 7.6 μg pMD2.G (Addgene, 12259), and 13.9 μg psPAX2 (Addgene, 12260) using 125 μl Mirus *Trans*IT-293 (Mirus, MIR 2700) in Opti-MEM (Gibco, 31985062). Lentivirus-containing supernatant was harvested two days post-transfection, filtered through a 0.45 μm PVDF membrane (Millipore, SLHV033RS), and concentrated using precipitation solution (Alstem, VC100). To determine functional lentiviral titer, iPS cells were transduced in a dilution series and fluorophore-positive populations were quantified using a flow cytometer three days post-transduction.

### Human-specific deletions
hDel sequences (Supplementary Data 1–13[1]) were extracted from panTro6 contigs and aligned to the assembled panTro6 reference genome. hDel coordinates were compared to the UCSC hg38-panTro6 net alignment using BEDTools (v. 2.27.1) and discrepancies between hDels, hCONDELs[6], and the UCSC hg38-panTro6 net alignment were resolved by reciprocal BLAST-like alignment tool.

### hDel enrichment analysis
Intersections between hDels and genomic features were tested for significance using a resampling approach. After discarding all hDel sequences mapped to unplaced contigs and alternate haplotypes, 1000 matched null sets of genomic features were sampled from panTro6 (excluding unplaced contigs and alternate haplotypes) using bootRanges[55]. Regions were sampled using a block length of 500 kb, excluding approximately 51 Mb of blacklisted sequence (excludeOption = "drop", withinChrom = FALSE). This blacklist was generated by combining runs of at least 1 kb with less than 100% 50-mer mappability identified using GenMap[56] and nuclear mitochondrial insertions (NUMTs) inferred as described previously[57].

After sampling matched null features, intersection between each sampled set of features and the feature set of interest (introns, exons, repeat elements, Tn-5 accessible regions, and pA-Tn5-accessible

regions) was assessed using BEDTools[58]. After counting the number of base pairs in each intersection, *p*-values for enrichment or depletion of intersection were computed as the number of null feature sets for which intersection with the feature set of interest was greater (for enrichment testing) or lower (for depletion testing) than the observed intersection between hDels and features of interest.

Introns, exons, and intergenic regions were extracted from the chimpanzee reference genome (panTro6), annotated with the Comparative Annotation Toolkit[59,60]. Repeat regions were identified using the UCSC panTro6 RepeatMasker annotations. All features were merged using BEDTools before performing intersections.

Bases under selective constraint were computed using PhyloP[61] on an alignment of 241 mammal genomes[62], with panTro6 as the reference sequence. Bases with at least 95% probability of purifying selection were taken to be under selective constraint; this corresponded to 25.6% of all exonic bases and resulted in enrichment of constrained bases in exons ($p < 10^{-3}$, using the same resampling approach described above).

## hDel sgRNA library design

For all hDel sgRNA libraries (hDel-v1, hDel-v2, hDel-v3, and hDel-v4), candidate hDel-targeting sgRNAs were identified and scored for predicted off-target activity against mismatched target sites in the chimpanzee reference genome (panTro6) using FlashFry[63] (v. 1.15; –maxMismatch = 3; –scoringMetrics JostandSantos, dangerous, minot). Candidate sgRNAs ($GN_{19}$) with >1 perfect-match target site, CRISPRi specificity score <0.2, maximal predicted CRISPRi activity at any off-target site > 0.8, or TTTT sequences were excluded from all libraries.

To maximize coverage of hDel sequences in the hDel-v1 library, sgRNAs were grouped into non-overlapping 50-bp bins corresponding to the genomic location of their target sequence. Candidate sgRNAs were then scored for predicted on-target activity using DeepHF[18], and the sgRNA with the highest DeepHF score in each 50-bp bin was selected for inclusion in the library ($n = 3121$ hDels). For hDels targeted by fewer than 10 sgRNAs after filtering and binning, sgRNAs were ranked by their DeepHF scores and a sequentially increasing number of sgRNAs were selected per 50-bp bin until all hDels were targeted by at least 10 sgRNAs ($n = 1531$ hDels). For the remaining hDels targeted by fewer than 10 sgRNAs, sgRNAs targeting human-conserved sequence flanking either side of each hDel were filtered, binned, and ranked as described above until each hDel ± 250 bp was targeted by at least 5 sgRNAs ($n = 1706$ hDels). In total, 170,904 sgRNAs tiling 6358 hDels were included in hDel-v1 (median distance between sgRNAs: 52 bp; median number of sgRNAs per hDel: 14). Non-targeting sgRNAs ($n = 3000$ sgRNAs) were generated by scoring random $GN_{19}NGG$ sequences against panTro6 and filtering for 0 perfect-match target sites. As protein-coding controls, sgRNAs targeting the promoters of essential genes ($n = 8068$ sgRNAs targeting 2017 genes), proliferation-suppressor genes ($n = 1692$ sgRNAs targeting 423 genes), and chimpanzee organoid-expressed genes ($n = 15,189$ sgRNAs targeting 5063 genes) were selected from hCRISPRi-v2 after off-target scoring against panTro6. The complete hDel-v1 sgRNA library contains 198,718 sgRNAs. All hDel-v1 sgRNAs were screened in a single pool.

Candidate hDel *cis*-regulatory elements were screened at higher tiling density in the hDel-v2 sgRNA library ($n = 558$ hDels, see hDel CRISPRi screening analysis). For the 50 hDels with the greatest number of sgRNAs passing off-target filters described above, all sgRNAs within ± 500 bp of MAGeCK- and iAnalyzer-identified hDel genomic windows and DESeq2-identified singleton sgRNAs were selected for inclusion in the library. For all other hDels, all hDel-targeting sgRNAs, as well as sgRNAs targeting human-conserved sequence flanking either side of each hDel (± 500 bp), were included. In total, 78,270 sgRNAs tiling 558 hDels were included in hDel-v2 (median distance between sgRNAs: 7 bp; median number of sgRNAs per hDel: 119). Non-targeting sgRNAs

($n = 2000$ sgRNAs) and sgRNAs targeting the promoters of essential genes ($n = 600$ sgRNAs targeting 98 genes) and proliferation-suppressor genes ($n = 394$ sgRNAs targeting 60 genes) were selected from hDel-v1 and CEV-v1, respectively. The complete hDel-v2 sgRNA library contains 81,264 sgRNAs. All hDel-v2 sgRNAs were screened in a single pool.

Two additional sgRNA libraries were designed for single-cell CRISPRi screening to facilitate mapping hDel *cis*-regulatory element-gene pairs. For hDel-v3, all essential sgRNAs targeting α-RRA-identifed 250-bp hDel genomic windows ($n = 122$ sgRNAs targeting 19 hDels, see hDel CRISPRi screening analysis) were selected for inclusion in the library. sgRNAs targeting the promoters hDel-proximal genes ($n = 18$ sgRNAs targeting 12 genes), core-control non-targeting sgRNAs (Replogle *Cell* 2022) ($n = 10$ sgRNAs), and sgRNAs targeting putative *cis*-regulatory elements (Gasperini *Cell* 2019) ($n = 10$ sgRNAs targeting 5 *cis*-regulatory elements) were also included. The complete hDel-v3 sgRNA library contains 160 sgRNAs.

For hDel-v4, nonessential sgRNAs targeting hDels marked by chromatin state features associated with *cis*-regulatory elements, including Omni ATAC-seq, H3K4me1, and H3K27ac (see Omni ATAC-seq and CUT&Tag), were selected for inclusion in the library. sgRNAs passing off-target filters described above and lacking AAAA, TTTT, GGGG, or CCCC sequences were scored for predicted on-target activity using DeepHF, and the 5 sgRNAs with the highest DeepHF scores targeting each hDel Tn5- or pA-Tn5-accessible region were included ($n = 888$ sgRNAs targeting 163 hDels). Core-control non-targeting sgRNAs ($n = 25$ sgRNAs), and sgRNAs targeting putative *cis*-regulatory elements ($n = 10$ sgRNAs targeting 5 *cis*-regulatory elements) were also included. The complete hDel-v4 sgRNA library contains 923 sgRNAs.

## hDel sgRNA library cloning

Oligonucleotide pools were designed with flanking PCR adapter sequences and restriction sites (*BstXI*, *BlpI*), synthesized by Agilent Technologies, and cloned into the sgRNA expression vector pCRISPRia-v2 (Addgene, 84832; hDel-v1, hDel-v2) or pJR101 (Addgene, 187241; hDel-v3, hDel-v4) as described previously[54]. Briefly, oligonucleotide pools were amplified by 8 to 10 cycles of PCR using NEBNext Ultra II Q5 Master Mix (New England Biolabs, M0544X), digested with *BstXI* and *BlpI*, size-selected by polyacrylamide gel electrophoresis, ligated into *BstXI*- and *BlpI*-digested pCRISPRia-v2 or pJR101, and introduced into MegaX DH10B T1$^R$ cells by electroporation (Invitrogen, C640003; Bio-Rad, 1652660).

## hDel CRISPRi screening

Chimpanzee CRISPRi iPS cells (C3624K or Pt5-C, hDel-v1; C3624K, hDel-v2) were dissociated with Accutase (Innovative Cell Technologies, AT104-500), resuspended in v3.1 medium supplemented with 2 µM thiazovivin and 5 µg/ml polybrene (Mirus, MIR 6620), transduced with the hDel-v1 or hDel-v2 lentiviral sgRNA library at a target infection rate of 25%, and plated at a density of 85,000 cells/cm² in Matrigel-coated 5-layer cell culture flasks (Corning, 353144). Two days post-transduction, cells were dissociated with Accutase, resuspended in v3.1 medium supplemented with 2 µM thiazovivin and 1.5 µg/ml puromycin, and plated at a density of 100,000 cells/cm². Four days post-transduction, 200 M cells were harvested ($t_0$) and 300 M cells were resuspended in v3.1 medium supplemented with 1.5 µg/ml puromycin and plated (≥1000× sgRNA library representation). Selection efficiency was assessed using a flow cytometer (≥70% BFP + ). Every two days, cells were dissociated with Accutase, resuspended in v3.1 medium supplemented with 2 µM thiazovivin, and plated. Technical replicates were maintained separately for the duration of the screen. After 10 days of growth, 200 M cells from each technical replicate were harvested ($t_{final}$). Genomic DNA was isolated from pelleted cells by column purification (Macherey-Nagel, 740950.50), and the sgRNA

expression cassette was amplified by 22 cycles of PCR using NEBNext Ultra II Q5 Master Mix and primers containing Illumina P5/P7 termini and sample-specific TruSeq indices. Each sample was distributed into individual 100 µl reactions in 96-well plates, each containing 10 µg genomic DNA. Following amplification, reactions from each sample were pooled and a 100 µl aliquot was purified by double-sided SPRI selection (0.65×, 1×). Purified libraries were quantified using Agilent Bioanalyzer, pooled at equimolar concentrations, and sequenced on Illumina HiSeq 4000 using a custom sequencing primer (SE50; oCRISPRi_seq V5).

## hDel CRISPRi screening analysis

Single-end sequencing reads were aligned to hDel-v1 or hDel-v2 and counted using MAGeCK[64] (v. 0.5.9.4; count). For hDel-v1 (C3624K, Pt5-C), sgRNAs were assigned to overlapping 500-bp hDel genomic windows (250-bp step size) depending on the genomic location of their target sequence. hDel genomic windows targeted by fewer than 5 sgRNAs were excluded from analysis. hDel genomic windows and sgRNA counts were then used as input for analysis by MAGeCK alpha-robust rank aggregation (α-RRA) (test –paired –gene-test-fdr-threshold 0.05 –norm-method control –gene-lfc-method alphamean), iAnalyzeR (combining sgRNA $Z$-scores for each genomic window using Stouffer's method followed by the Benjamini–Hochberg procedure), or DESeq2 (design = -individual + time). The union of the hDel 500-bp genomic windows and hDel-targeting sgRNAs identified using these approaches ($n = 313$ genomic windows MAGeCK α-RRA 10% FDR, $n = 147$ genomic windows iAnalyzeR 5% FDR, $n = 87$ genomic windows MAGeCK α-RRA 10% FDR and iAnalyzeR 5% FDR, $n = 202$ sgRNAs DESeq2 1% FDR) were included in the hDel-v2 sgRNA library ($n = 558$ hDels). For the hDel-v1 Manhattan plot in Fig. 1, sgRNA adjusted $p$-values from DESeq2 were combined into FDRs corresponding to each 500-bp hDel genomic window using α-RRA (v. 0.5.9; –control).

For hDel-v2 (C3624K), sgRNA counts were used as input for differential analysis by DESeq2 (design = -time). sgRNAs containing GGGG sequences following spacer position 5 or CCCC sequences between spacer positions 10 and 12 were excluded from analysis due to pervasive off-target effects (Supplementary Fig. 7; fraction of significantly enriched or depleted sgRNAs at least twofold greater than all hDel-v2 sgRNAs). Enriched sgRNAs (log$_2$ fold-change ≥1) initially present at low abundance (≤5th percentile) were also excluded from analysis ($n = 159$ sgRNAs). hDel-targeting sgRNAs were assigned to nonoverlapping 250-bp hDel genomic windows, and sgRNA adjusted $p$-values from DESeq2 were combined into FDRs corresponding to each hDel genomic window using α-RRA (–control).

## Single-cell hDel CRISPRi screening

For hDel-v3, chimpanzee CRISPRi C3624K iPS cells were dissociated with Accutase (Innovative Cell Technologies, AT104-500), resuspended in v3.1 medium supplemented with 2 µM thiazovivin and 5 µg/ml polybrene (Mirus, MIR 6620), transduced with the hDel-v3 lentiviral sgRNA library at a target infection rate of 10%, and plated at a density of 100,000 cells/cm². The following day, v3.1 medium was replaced. Two days post transduction, v3.1 medium was replaced and supplemented with 1.5 µg/ml puromycin. Selection continued until six days post transduction. Seven days post transduction, iPS cells were dissociated with Accutase and resuspended in 1× PBS supplemented with 0.04% BSA for single-cell RNA sequencing (Direct-capture Perturb-seq). iPS cells were partitioned into Gel Beads-in-emulsion (GEMs) across five wells using the 10× Genomics Chromium Controller and cDNA libraries from polyadenylated mRNAs and Feature Barcode-compatible sgRNAs were generated by following the 10× Genomics Chromium Next GEM Single Cell 3′ Reagent Kits v3.1 (Dual Index) User Guide (CG000316 Rev D). cDNA libraries from mRNAs and sgRNAs were quantified using Agilent Bioanalyzer, pooled

at a 4:1 molar ratio, and sequenced on Illumina NovaSeq 6000 (28 × 10 × 10 × 90).

For hDel-v4, chimpanzee CRISPRi C3624K iPS cells were transduced with the hDel-v4 lentiviral sgRNA library at a target infection rate of >95% and Direct-capture Perturb-seq was performed as described above.

## Single-cell CRISPRi screening analysis

A mismatch map for hDel-v3 and hDel-v4 was generated using kITE[65] and indexed using kallisto[66] (v. 0.48.0; kb ref –workflow kite). Paired-end sequencing reads from the gene expression and CRISPR screening libraries were then pseudoaligned and error-collapsed using kallisto | bustools (v. 0.42.0; kb count –workflow kite:10 × FB –filter bustools). The chimpanzee reference genome (panTro6), annotated with the Comparative Annotation Toolkit[59,60] was used as the reference transcriptome.

To assign sgRNAs to cells, a two-component Poisson-Gaussian mixture model[67] was fit for each sgRNA using the log$_2$-transformed UMIs in the bustools-filtered cell by sgRNA matrix. Assignments were made when the posterior probability of a cell belonging to the second component of the mixture model was >0.5.

For hDel-v3, cells with fewer than 2049 genes detected (10th percentile), 5105 UMIs (10th percentile), and greater than 15% mitochondrial UMIs were filtered from the dataset using Scanpy (v. 1.9.1). After intersecting Cell Barcodes in the gene expression and CRISPR screening UMI matrices, 16,810 cells were retained for analysis (median 4564 genes detected per cell; median 15,366 gene UMIs per cell; median 151 cells per sgRNA; median 1597 UMIs per sgRNA per cell).

For hDel-v4, cells with fewer than 1106 genes detected (10th percentile), 2744 UMIs (10th percentile), and less than 1% or greater than 15% mitochondrial UMIs were filtered from the dataset. After intersecting Cell Barcodes in the gene expression and CRISPR screening UMI matrices, 18,571 cells were retained for analysis (median 6143 genes detected per cell; median 26,989 gene UMIs per cell; median 7 sgRNAs per cell; median 195 cells per sgRNA; median 495 UMIs per sgRNA per cell).

To perform differential gene expression testing for hDel-v3, the unnormalized gene expression UMIs were summed across cells containing each sgRNA (+sgRNA "pseudobulk") and cells containing non-targeting sgRNAs (+non-targeting sgRNA "pseudobulk") using ADPBulk (https://github.com/noamteyssier/adpbulk) and a likelihood-ratio test was performed individually for each sgRNA using DESeq2 controlling for GEM well (design = -GEM+sgRNA; test = "LRT", reduced = -GEM).

To perform differential gene expression testing for hDel-v4, the unnormalized gene expression UMIs were summed across cells containing each sgRNA (+sgRNA "pseudobulk") and all other cells (-sgRNA "pseudobulk") using ADPBulk and a likelihood-ratio test was performed individually for each sgRNA using DESeq2 controlling for GEM well (design = -GEM + sgRNA; test = "LRT", reduced = -GEM). Cells containing any sgRNA targeting the same hDel were excluded from the -sgRNA "pseudobulk".

For hDel-v3 and hDel-v4, genes detected in fewer than 2000 cells (-10% of all cells) were filtered from the dataset prior to differential gene expression testing. To identify differentially expressed genes for hDel-targeting sgRNAs, log$_2$ fold-change was divided by its standard error for each sgRNA-gene pair and then $Z$-score normalized. Gene-specific null distributions were generated from all sgRNA-gene pairs located ≥100 kb from a hDel. For genes within ≤100 kb of a hDel, $p$-values were computed from the survival function of a normal distribution, concatenated across genes, and adjusted for multiple comparisons using the Benjamini–Hochberg procedure applied to sgRNA-gene pairs within ≤100 kb.

## hDel Cas9 RNP nucleofection

Chimpanzee iPS cells from a male donor (C8861) were cultured in v3.1 iPSC medium supplemented with 5 mg/ml BSA at 37 °C and 5% $CO_2$, as described above. Cells were dissociated with Accutase (Sigma-Aldrich, A6964-500ML), resuspended in P3 solution (Lonza, V4XP-3024), and nucleofected using an Amaxa 4D Nucleofector (CA-137). For each hDel, pairs of RNPs were prepared by complexing sgRNAs (Synthego) with Alt-R S.p. Cas9 Nuclease V3 (IDT, 1081058) and incubating for 15′ at room temperature. Per cuvette, $1 \times 10^6$ cells were electroporated with Cas9 RNPs and pmaxGFP (Lonza). For hDel_2247, crRNA sequences ATGGAGAT GTTGTCGGGCA and GGAACTCTGGGTTTGCCAG were used. For hDel_ 3779, cRNA sequences TGCCTCCAGTGGTAGAGTAG and TCCCCACTA AGACAGAGAAA were used. Matched no-sgRNA controls were nucleofected in parallel with sgRNAs omitted. Nucleofected cells were plated at ~104,000 cells/cm² on Matrigel-coated 6-well plates. At ~90% confluency, cells were dissociated with PBS containing 0.5 mM EDTA and harvested for analysis. For each hDel, three independent nucleofections and two independent no-sgRNA control nucleofections were performed.

## hDel genotyping

Genomic DNA was extracted using the Monarch Genomic DNA Purification Kit (NEB, T3010). Wild-type chimpanzee (C4933) and human (H20961) genomic DNA were included as controls. Reactions contained 0.4 μM forward and reverse primers (hDel_2247 forward: GGTTTAGTATCCAAGACTCCTAT; hDel_2247 reverse: GGAAGGGTAA ACAGACTATAGTC; hDel_3779 forward: TTCCTATTTCGAGGTGGGTC AGC; hDel_3779 reverse: CTGAAACAGTTGTTGTGGAGGCC), 1× Long-Amp Taq 2× Master Mix (New England Biolabs, M0287S), and genomic DNA in nuclease-free water. For hDel_2247, 50 ng DNA was used with the following cycling conditions: 94 °C for 30 s; 30 cycles of 94 °C for 20 s, 52 °C for 30 s, 65 °C for 9 min 20 s; final extension at 65 °C for 10 min. Products were resolved by agarose gel electrophoresis. For hDel_3779, 25 ng DNA was used with the following cycling conditions: 94 °C for 30 s; 30 cycles of 94 °C for 15 s, 61 °C for 20 s, 65 °C for 1 min 40 s; final extension at 65 °C for 10 min. Products were resolved by agarose gel electrophoresis.

## Reverse transcription quantitative PCR

Total RNA was extracted using the Monarch Total RNA Miniprep Kit (New England Biolabs, T2010). For each sample, 1 μg RNA was used as input for cDNA synthesis (Thermo Fisher, K1671) using the following conditions: 25 °C for 10 min, 50 °C for 60 min, 85 °C for 5 min. RT-qPCR included 1× PowerUp SYBR Green Master Mix (Applied Biosystems, A25742), 0.25 μM forward and reverse primers, cDNA diluted 1:80, and nuclease-free water. For each hDel, three primer pairs were used: *GAPDH* and *POLR2C* (reference genes), and *PLPP1* or *FAM49B* (target genes). Reactions were run in quadruplicate in 10 μl volumes on 384-well plates. RT-qPCR was performed on QuantStudio 6 Real-Time PCR System (Applied Biosystems) using the Relative Quantification module in Fast mode with a heated cover at 105 °C. Cycling included a hold stage (50 °C for 2 min, 95 °C for 2 min), followed by 40 cycles (95 °C for 15 s, 55 °C for 20 s, 60 °C for 30 s) with fluorescence collected at each step. Melt curve analysis (95 °C for 15 s, 55 °C for 20 s, 95 °C for 15 s) was performed with continuous acquisition. Cq values were calculated using the ΔΔCT method with *GAPDH* and *POLR2C* as reference genes. Technical replicates were averaged, and relative expression (RQ) was calculated per sample and group. Pairwise comparisons were tested using two-sided and one-sided Student's *t*-tests to assess both general and directional effects. One-way ANOVA with Tukey's HSD post-hoc test was used to detect group-wide differences while correcting for multiple comparisons.

## Omni ATAC-seq

Chimpanzee iPS cells from four healthy donors (C3624K dCas9-KRAB, Pt5-C dCas9-KRAB, C8861, and C3651) were rinsed with DMEM/Ham's F-12 and dissociated with PBS supplemented with 0.5 mM EDTA. Cells were washed and resuspended in cold PBS supplemented with 0.04% BSA. Following counting, 100,000 cells were resuspended in 100 μl cold lysis buffer (10 mM Tris-HCl pH 7.4, 10 mM NaCl, 3 mM $MgCl_2$, 0.01% digitonin, 0.1% Tween-20, 0.1% NP40, and 1% BSA) by pipetting three times and incubated on ice for 3 min. Following lysis, 1 ml cold wash buffer (10 mM Tris-HCl pH 7.4, 10 mM NaCl, 3 mM $MgCl_2$, 0.1% Tween-20, and 1% BSA) was added and nuclei were centrifuged at $500 \times g$ for 5 min at 4 °C. Supernatant was removed using P1000, P200, and P20 pipettes, and nuclei were resuspended in 50 μl transposition mix (25 μl 2× transposition buffer (Active Motif), 10 μl transposase (Active Motif), 2 μl 10× PBS, 0.5 μl 0.5% digitonin, 0.5 μl 10% Tween-20, and 12 μl dH2O) by pipetting three times. Transposition reactions were incubated for 30 min at 37 °C while shaking at 1000 rpm and cleaned using DNA Clean & Concentrator-5 (Zymo Research, D4003). Transposed libraries were amplified in 50 μl PCRs (0.5 μl Q5 DNA polymerase, 10 μl 5× Q5 reaction buffer, 1 μl 10 mM dNTP mix, 2.5 μl 25 μM i7 index primer and 2.5 μl 25 μM i5 index primer[68]) using the following cycling parameters: 72 °C for 5 min; 98 °C for 30 s; 10 cycles of 98 °C for 10 s, 63 °C for 30 s, and 72 °C for 1 min. Amplified transposed libraries were purified by SPRI selection (1.2×), quantified using Agilent Bioanalyzer, and pooled at equimolar concentrations. Purified libraries were sequenced on Illumina HiSeq 4000 (PE100).

## Neuroepithelial cell differentiation and Omni ATAC-seq

Chimpanzee (C3624K dCas9-KRAB, C8861), human (H20961), and orangutan (Jos-3C1) iPS cells were seeded at a density of 100,000 cells/ cm² in v4 medium (v3.1 medium with 5 mg/ml BSA) containing CEPT[69]. The following day (Day 1), medium was replaced with DMEM/Ham's F-12 differentiation medium (DMEM/Ham's F-12 containing 1% polyvinyl alcohol, 100 μg/ml 2-Phospho-L-ascorbic acid trisodium salt, 20 ng/ml sodium selenite, 20 μg/ml holo-transferrin, 20 μg/ml insulin, and 0.1 mg/ml Primocin) supplemented with 0.5 μM LDN193189, 10 μM SB431542, and 0.1 μM Wnt-C59. Medium was replaced on Day 3. On Day 5, medium was replaced with DMEM/Ham's F-12 differentiation medium supplemented with 0.5 μM LDN193189, 10 μM SB431542, and CEPT. On Day 7, cells were dissociated with Accutase and seeded at a density of 800,000 cells/cm² onto polyethylenimine (0.1%) and Matrigel-coated plates in DMEM/Ham's F-12 differentiation medium supplemented with a modified 6F[70] formulation (1 μM SB431542, 0.1 μM K02288, 0.1 μM AKTiVIII, 0.075 μM MK2006, 0.1 μM LDN193189, 0.5 μM CHIR99021, 0.2 μM NVP-TNKS656, 25 ng/ml SHH), and CEPT. On Day 10, medium was replaced with DMEM/Ham's F-12 differentiation medium supplemented with 25 ng/ml SHH, 50 ng/ml FGF8, and CEPT. On Day 12, medium was replaced with DMEM/Ham's F-12 differentiation medium supplemented with 25 ng/ml SHH, and 50 ng/ml FGF8. On Day 15, medium was replaced and cells were rinsed with DMEM/Ham's F-12 and dissociated with Accutase. Omni ATAC-seq was performed as described above. Purified libraries were sequenced on Illumina NovaSeq X (PE150).

## Omni ATAC-seq analysis

Paired-end sequencing reads were trimmed using Cutadapt[71] (v. 3.4; -q 20 –minimum-length 20) and aligned to panTro6 using Bowtie 2[72] (v. 2.2.5; –very-sensitive -X 2000 -k 10). SAM files were converted to BAM format while discarding alignments with MAPQ < 15, sorted by position or read name, and indexed using SAMtools[73] (v. 1.10, -q 15). Tn5-accessible regions were called following PCR duplicate removal using Genrich (https://github.com/jsh58/Genrich) (v. 0.6.1; -j -r -e chrM -q 0.05). Tn5-accessible regions were intersected using BEDTools multiinter (v. 2.27.1) and only regions common to all four iPS cell lines were retained for analysis (n = 94,003). BAM files were converted to bedGraph format using deepTools[74] (v. 3.5.1; –binSize 1 –ignoreDuplicates) and visualized using SparK (https://github.com/harbourlab/SparK) (v. 2.6.2). To identify intersections between Omni ATAC-seq and hDels, at least half of a Tn5-accessible region was required to intersect a hDel.

## CUT&Tag

Concanavalin A-coated beads (Epicypher, 21-1401) were washed twice and resuspended in binding buffer (20 mM HEPES pH 7.5, 10 mM KCl, 1 mM $CaCl_2$, and 1 mM $MnCl_2$). Chimpanzee iPS cells from two healthy donors (C3624K dCas9-KRAB and C8861) were rinsed with DMEM/Ham's F-12 and dissociated with PBS supplemented with 0.5 mM EDTA. Following counting, 175,000 cells were washed twice in 1 ml wash buffer (20 mM HEPES pH 7.5, 150 mM NaCl, 0.5 mM spermidine, and 1 tablet cOmplete EDTA-free protease inhibitor cocktail) and incubated with 11 µl concanavalin A-coated beads on an end-over-end rotator for 10 min. The cell and bead mixture was placed on a magnetic stand and unbound supernatant was discarded. Bead-bound cells were resuspended in 50 µl primary antibody buffer (20 mM HEPES pH 7.5, 150 mM NaCl, 0.5 mM spermidine, 1 tablet cOmplete EDTA-free protease inhibitor cocktail, 0.05% digitonin, 2 mM EDTA, and 0.1% BSA) and 0.5 µl primary antibody (1:100 dilution; H3K4me1 (Abcam, ab8895, lot GR3426435-2), H3K4me3 (Abcam, ab8580, lot GR3425199-1), H3K27ac (Abcam, ab4729, lot GR3442878-1), or H3K27me3 (Cell Signaling Technology, 9733S, lot 19)) was added. Bead-bound cells and primary antibody were mixed by pipetting and placed on a nutator overnight at 4 °C. The following day, the primary antibody solution was placed on a magnetic stand and supernatant was discarded. Bead-bound cells were resuspended in 50 µl secondary antibody buffer (goat anti-rabbit IgG (Epicypher, 13-0047) diluted 1:100 in 20 mM HEPES pH 7.5, 150 mM NaCl, 0.5 mM spermidine, 1 tablet cOmplete EDTA-free protease inhibitor cocktail, and 0.05% digitonin) and placed on a nutator at room temperature for 60 min. Bead-bound cells were washed three times in 1 ml digitonin-wash buffer (20 mM HEPES pH 7.5, 150 mM NaCl, 0.5 mM spermidine, 1 tablet cOmplete EDTA-free protease inhibitor cocktail, and 0.05% digitonin), resuspended in 50 µl digitonin-300 buffer (20 mM HEPES pH 7.5, 300 mM NaCl, 0.5 mM spermidine, 1 tablet cOmplete EDTA-free protease inhibitor cocktail, and 0.05% digitonin) containing pAG-Tn5 (Epicypher, 15-1017), and placed on a nutator at room temperature for 60 min. Following pA-Tn5 binding, bead-bound cells were washed three times in 1 ml digitonin-300 buffer, resuspended in 125 µl transposition buffer (10 mM $MgCl_2$ in digitonin-300 buffer), and incubated at 37 °C for 60 min. Following transposition, 4.2 µl 0.5 M EDTA, 1.25 µl 10% SDS, and 1.1 µl 20 mg/ml proteinase K (Invitrogen, AM2546) were added and bead-bound cells were incubated at 55 °C for 60 min. Bead-bound cells were placed on a magnetic stand and DNA-containing supernatant was cleaned using ChIP DNA Clean & Concentrator (Zymo Research, D5201). Transposed libraries were amplified in 50 µl PCRs (25 µl NEBNext HiFi 2× PCR master mix, 2 µl 10 µM i7 index primer and 2 µl 10 µM i5 index primer[68]) using the following cycling parameters: 58 °C for 5 min; 72 °C for 5 min; 98 °C for 45 s; 13 cycles of 98 °C for 15 s, 60 °C for 10 s; and 72 °C for 1 min. Amplified transposed libraries were purified by SPRI selection (1.3×), quantified using Agilent Bioanalyzer, and pooled at equimolar concentrations. Purified libraries were sequenced on Illumina NovaSeq 6000 (PE150).

## CUT&Tag analysis

Paired-end sequencing reads were trimmed using Cutadapt (-q 20 --minimum-length 20) and aligned to panTro6 using Bowtie 2 (--very-sensitive -X 700 -k 10). SAM files were converted to BAM format while discarding alignments with MAPQ < 15, sorted by position or read name, and indexed using SAMtools (-q 15). pA-Tn5-accessible regions were called following PCR duplicate removal using Genrich (-j -r -e chrM -q 0.05). For each primary antibody, pA-Tn5-accessible regions were intersected using BEDTools intersect and only regions common to both iPS cell lines were retained for analysis. BAM files were converted to bedGraph format using deepTools (--binSize 1 --ignoreDuplicates) and visualized using SparK. To identify intersections between CUT&Tag and hDels, at least half of a pA-Tn5-accessible region was required to intersect a hDel.

## Mouse ENCODE analysis

Mouse ENCODE ATAC-seq and H3K27ac and p300 ChIP-seq datasets from E14.5 forebrain, limb, heart, lung, and liver tissues were downloaded from (https://www.encodeproject.org/). For all datasets, the bigWig file "fold-change over control, isogenic replicates 1,2" was used. bigWig files were converted to bedGraph format using UCSC bigWigToBedGraph and visualized using SparK.

## snATAC-seq of PCD80 rhesus macaque prefrontal cortex

Prefrontal cortex (PFC) was microdissected from post-conception day 80 (PCD80) rhesus macaque cortex[75]. Tissue was enzymatically dissociated in papain (Worthington, LK003176) containing DNase I for 30 min at 37 °C and gently triturated to form a single-cell suspension. Cells were washed and resuspended in cold PBS supplemented with 0.04% BSA. Following counting, 100,000 cells were resuspended in 100 µl cold lysis buffer (10 mM Tris-HCl pH 7.4, 10 mM NaCl, 3 mM $MgCl_2$, 0.01% digitonin, 0.1% Tween-20, 0.1% NP40, and 1% BSA) by pipetting three times and incubated on ice for 3 min. Following lysis, 1 ml cold wash buffer (10 mM Tris-HCl pH 7.4, 10 mM NaCl, 3 mM $MgCl_2$, 0.1% Tween-20, and 1% BSA) was added and nuclei were centrifuged at 500 × $g$ for 5 min at 4 °C. Nuclei were partitioned into GEMs using the 10× Genomics Chromium Controller and transposed libraries were generated by following the 10× Genomics Chromium Chromium Next GEM Single Cell ATAC Reagent Kits v1.1 User Guide (CG000209 Rev G). Amplified transposed libraries were quantified using Agilent Bioanalyzer and sequenced on Illumina NovaSeq 6000 (51 × 12 × 24 × 51).

## snATAC-seq analysis

Paired-end sequencing reads were aligned to rheMac10 (BSgenome.Mmulatta.UCSC.rheMac10, TxDb.Mmulatta.UCSC.rheMac10.refGene) using Cell Ranger ATAC (v. 2.1.0) and processed with ArchR[76] (v. 1.0.2). Cells with fewer than 3000 fragments or a TSS enrichment score less than 20 were filtered from the dataset ($n = 5166$ cells retained, median 9142 fragments per cell, median 31.8 TSS enrichment score). Cellular subsets were identified by iterative latent semantic indexing dimensionality reduction followed by graph-based clustering using ArchR. Fragments were then summed across cells within each cluster using pycisTopic[77] (v. 1.0.3) and visualized using SparK.

## Transgenic *lacZ* reporter assay

The chimpanzee sequence contained within hDel_2247 intersecting Omni ATAC-seq, H3K4me1, and H3K27ac (1557 bp; chr5:45194159-45195715, panTro6) was synthesized and cloned into the *lacZ* reporter vector *Hsp68_mini-LacZ-SV40* (VectorBuilder). Transient transgenic mice were produced by pronuclear injections and analyzed for *lacZ* activity at embryonic day 13 (E13.5) (Cyagen Biosciences).

## Data availability

Paired-end sequencing reads (FASTQ) generated in this study have been deposited on SRA under BioProject PRJNA1002791. Source data for all figures is supplied as a source data file. Source data are provided with this paper.

## Code availability

Notebooks implementing analyses are available at https://github.com/tdfair.

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

## Acknowledgements

We thank Nadav Ahituv, Jingwen Ding, Luke Gilbert, Max Haeussler, Craig Lowe, Aaron McKenna, Caroline Mrejen, Michael Mui, Katie Pollard, Joseph Replogle, Demian Sainz, Noam Teyssier, Lee Spraggon at the READY Center, and members of the Pollen group for valuable discussions. This work was supported by the following funding sources: Ruth L. Kirschstein National Research Service Predoctoral Fellowship Award F31 HG011569-01A1 (T.F.), Weill Neurohub Fellowship (N.K.S.), National Institutes of Health DP2MH122400-01, Schmidt Futures Foundation, Shurl and Kay Curci Foundation Innovative Genomics Institute Award. A.A.P. is a New York Stem Cell Foundation Robertson Investigator. This project was funded in part by the Emory National Primate Research Center Grant No. ORIP/OD P51OD011132.

## Author contributions

T.F. performed experiments and analyzed data. B.J.P. performed neuroepithelial cell differentiations. D.S. and O.E.C. performed Cas9 RNP validation experiments and analyzed data. N.K.S. performed enrichment analyses. T.F. and A.A.P. conceived the study and wrote the manuscript with input from all authors.

## Competing interests

The authors declare no competing interests.
