## [Transparent Peer Review file · Nature Communications]

Mapping cis- and trans-regulatory target genes of human-specific deletions

Corresponding Author: Dr Alex Pollen

Version 0:

Reviewer comments:

Reviewer #1

(Remarks to the Author)

The manuscript by Fair et al describes their systematic functional tests of human-specific deletions (hDels) on cellular phenotypes using chimpanzee induced pluripotent stem cells (iPSCs). To my knowledge, this represents the first such study focusing on hDels and includes an impressive array of functional genomic experiments. Positing that they may contribute to human-specific traits, they compile a list of hDels via comparisons of human (GRCh38) and chimpanzee (panTro6) reference genomes, building from work by Kronenberg et al that used data from multiple apes to narrow in on these regions. Authors do not find overwhelming evidence that hDels play a major role in iPSC proliferation, using CRISPRi to silence hDel regions, but they are able to describe a handful of interesting examples. In some cases, they also identify target genes using Perturb-seq and connect with species' differentially-expressed (DE) genes from RNA-seq of chimpanzee-human hybrid cells. Though additional follow-up work is ultimately necessary to verify functional findings, we are left with preliminary evidence that hDels represent an understudied source of species' divergence. The manuscript is straightforward, well-written, and easy to follow. The experiments are comprehensive, well-designed/controlled, and additionally provide a useful genomic resource of chimpanzee iPSCs (e.g., ATAC-seq, RNA-seq, CUT&TAG).

Below are general comments aimed at improving the generalizability of findings and understanding if hDels might contribute to global human-specific traits:

1. Human-accelerated regions (HARs) and human-specific duplications (HSDs) are enriched for genes important in neural development. Alternatively, in this study, functional hDels seem to disproportionately impact ubiquitously expressed genes with innate cellular functions. Can authors comment on whether this is a product of the assay or a feature of hDels? One idea might be to take nearby associated genes of hDels and test for functional enrichment across all vs. hDels with identified functional impacts.
2. Further, since we have a set of functional hDels, are there any general rules that can be extracted from their sequence/genomic features? E.g., hConDels vs. non-conserved. Or proximity to genes. Currently, the study narrows in on a few interesting examples, but ideally, generalizable rules of hDels would be useful to the community.
3. Do authors have details on the population prevalence of identified hDels, particularly the ones they suggest are functional? In Kronenberg, et al they narrow in on ~5000 "fixed" fhDels, but here the study focuses on >7000. It's not clear where the additional 2000 comes from (polymorphic?). Are fixed hDels more likely to show a function or have defining genomic features?
4. In the hDel-v3 Perturb-seq, how do authors account for the possible biases introduced by pseudo-bulking across cells when testing functional impacts of hDels impacting proliferation? Might technical artefacts arise from genes with fewer cells (reduced proliferation) vs. more cells (increased proliferation), resulting in systematically fewer transcripts per gene after summing.

Minor comment:

1. For hDel-v3 Perturb-seq, only DE genes within 100 kbp of hDel are considered a target. How many hDel/gene combinations fit this criterion in the assay and was this factored into the model when determining statistical significance?

Reviewer #2

(Remarks to the Author)

Fair et al developed multiple CRISPRi screening libraries to investigate functions of human-specific deletions during evolution. Using both bulk and single-cell CRISPRi screenings, multiple human-specific deletions either affect cell proliferation in chimpanzee stem cells, or regulate cell gene expression. Studying these regions may reveal how non-coding regions play a role during evolution. In general, the idea is interesting and the approaches are sound. However, the research lacks key validation experiments to evaluate the overall quality of the screening and related conclusions. The authors should also proofread their manuscript, as several main figures were mislabeled.

Major

1. Related to sgRNA and hDel enrichment calculations in Figure 1, it is unclear what is the relationship between DESeq2 analysis on sgRNA and RRA analysis on hDel region enrichment. Would both data were used in later analysis? If a 250-b step size is used, some sgRNAs may be counted multiple times, would this affect the analysis?
2. It is unclear about the differences between V1 and V2 tiling. The author need to better describe it. Is the V2 library targeting the hits from V1?
3. Related to Figure 2c, how is it possible to have 18,148 sgRNAs in the scatter plot if the V2 library only contains 78,270 sgRNAs in total? What are compared?
4. If G4 and C4 would introduce strong off-target effects, why did not exclude these sgRNAs from V1 analysis? Furthermore, experimental evidence would be needed to show these sgRNAs introduce strong off-target effects, for instance, using discovery seq or other methods.
5. For the V4 analysis, there are multiple different sgRNAs per cell. The authors need to clarify how they correct the effects of these different sgRNAs in individual cells during single-cell RNA-seq analysis.
6. For the data related to hDel_2247, why mouse tissues were used for the comparison? What would be the results if human tissues were used?
7. There are no individual cell clones made to validate the screening results related to different phenotypes. As genetics screenings would suffer from false positive hits, these validations are key to show the screenings indeed identified proper hits and deletion regions.
8. The research aims to study genomic deletions in evolution, the authors would need to use CRISPR editing to delete at least some of the top hits to show the functions of the regions they identified and associated with different genes or phenotypes. Furthermore, non-coding regulatory regions, i.e. enhancer or silencer, may have different functions. CRISPRa may be needed to identify certain types of non-coding deletions, especially loss of silencer activity.
9. As the authors also agreed that the CRISPRi system has certain effector size limitations, how do the authors consider such an effect in their design of the smaller scale libraries, such as V2, V3, and V4?
10. The authors concluded that most hDels are dispensable for proliferation. Would this be a strong statement, as the screenings were only done in one cell type?

Minor

I am not sure if such mistakes are just minor issues, but a few figures were not labeled properly.

Figure 1 b and c were not referred to or referred wrongly.

Figure 2 panels were wrongly labeled.

Reviewer #3

(Remarks to the Author)

The manuscript by Fair et al., delves into a pivotal yet underexplored domain within genomics, specifically focusing on human-specific deletions (hDels) and their consequential roles in cellular processes. The study's innovative approach, utilizing cell proliferation and transcriptome changes as a functional readout, provides a fresh perspective in understanding these genomic elements. The authors' application of high throughput CRISPRi screening, executed both at bulk and single-cell levels, stands out as a methodologically robust technique to characterize the role of hDels. Through this screening, the identification of several hDels that influence cell proliferation or gene expression is a significant contribution to the field. Notably, the screening assays have been meticulously designed and executed, demonstrating a high level of scientific rigor. The data are presented with clarity and precision, making the study's findings accessible and comprehensible. The potential implications of these findings in the broader context of genomic research are substantial, indicating that this study, upon addressing specific concerns outlined below, merits timely publication in a prestigious academic journal.

Major Concerns:

1. Validation of hDels Function: The identified hDels from CRISPRi screen require further validation. It is recommended that region-specific deletions be executed using Cas9. This approach would solidify the functional attributions made to these hDels. Alternatively, this concern can also be addressed by analyzing the gene regulatory effects of cis-regulatory hDels in chimpanzee autotetraploids and allotetraploid cells by comparing gene expression between chimpanzee vs human alleles. Such comparative analysis would enhance our understanding of hDels' regulatory function/mechanisms when the sequences are deleted across species rather than epigenetically silenced.
2. For hDels identified from Perturb-seq and acting as candidate cis-regulatory elements (cCREs), what are the linear distance between these hDels and their target genes? Do these hDels function as distal cCREs? If so, does chromatin looping data in iPSC or hESC support such distal regulation? Given the availability of scRNA-seq data, analyses focusing on both local and distal regulation would deepen our understanding of hDel functions.
3. The 6.8 kb intergenic deletion hDel_2247 necessitates further elucidation. Since the typical effect size per gRNA in CRISPRi is usually less than 1kb, additional details characterizing the influence of sgRNAs on this larger region are needed. Are there specific motifs or subregions within this 6.8kb area that are critical for its function?

Minor Concern:

4. The text within several figures is too small to be easily readable.
5. There appears to be a discrepancy between the data presented in the figures and the conclusions drawn in the text. For instance, supporting data for the findings discussed in sections such as "We next examined RPL26 MRPS14, and MBD3..." and "sgRNAs targeting a putative inhibitor of DNA binding ID1 cis-regulatory element reduced the expression of ID1..." are not evident. Please include and clearly cite relevant figures or supplementary figures to substantiate these conclusions.

Version 1:

Reviewer comments:

Reviewer #1

(Remarks to the Author)

Authors have suitably addressed my comments and concerns. I look forward to seeing this exciting work published

Reviewer #2

(Remarks to the Author)

It is very common to validate some of the top hits from most of the genetic screenings to evaluate the performance of the screening, as both referees 2 and 3 pointed out. Especially that the authors would like to use CRISPRi to study the effects of genomic deletions. Related to that, do the 16 hDels identified from V1 screening overlap with the 38 hDels from V2?

It is also puzzling to see CRISPRi (with repressive activity) targeting a silencer (which probably regulated by a repressive TF) will lead to transcriptional activation, as showed by dDel_1608 and C4orf48.

Reviewer #3

(Remarks to the Author)

My concerns have been fully addressed. I recommend publication of the revised manuscript.

Version 2:

Reviewer comments:

Reviewer #2

(Remarks to the Author)

Related to comment #1, in the response letter, it indicates "Similarly, we find that deletion of hDel_3779 reduces the expression of FAM49B 9.1% to 27.6%." However, in the main text, it says "reduced the expression of the RAC1 effector FAM49B (Extended Data Fig. 8h, 8j, 14.2 to 23.6%, FDR < 0.1)". I am not sure if this is an error, but the authors should clarify this in the manuscript.

Related to comment #3, the author may need to include the references related to lncRNAs and silencers in the manuscript.

RESPONSE TO REVIEWERS' COMMENTS

Reviewer #1 (Remarks to the Author):

The manuscript by Fair et al describes their systematic functional tests of human-specific deletions (hDels) on cellular phenotypes using chimpanzee induced pluripotent stem cells (iPSCs). To my knowledge, this represents the first such study focusing on hDels and includes an impressive array of functional genomic experiments. Positing that they may contribute to human-specific traits, they compile a list of hDels via comparisons of human (GRCh38) and chimpanzee (panTro6) reference genomes, building from work by Kronenberg et al that used data from multiple apes to narrow in on these regions. Authors do not find overwhelming evidence that hDels play a major role in iPSC proliferation, using CRISPRi to silence hDel regions, but they are able to describe a handful of interesting examples. In some cases, they also identify target genes using Perturb-seq and connect with species' differentially-expressed (DE) genes from RNA-seq of chimpanzee-human hybrid cells. Though additional follow-up work is ultimately necessary to verify functional findings, we are left with preliminary evidence that hDels represent an understudied source of species' divergence. The manuscript is straightforward, well-written, and easy to follow. The experiments are comprehensive, well-designed/controlled, and additionally provide a useful genomic resource of chimpanzee iPSCs (e.g., ATAC-seq, RNA-seq, CUT&TAG).

Below are general comments aimed at improving the generalizability of findings and understanding if hDels might contribute to global human-specific traits:

1. Human-accelerated regions (HARs) and human-specific duplications (HSDs) are enriched for genes important in neural development. **Alternatively, in this study, functional hDels seem to disproportionately impact ubiquitously expressed genes with innate cellular functions.** Can authors comment on whether this is a product of the assay or a feature of hDels? One idea might be to take nearby associated genes of hDels and test for functional enrichment across all vs. hDels with identified functional impacts.

We thank Reviewer 1 for suggesting testing for enrichment of specific categories of genes near hDels, as well as near hDels that we found to contain *cis*-regulatory sequence. This analysis allowed us to include the latest Zoonomia conservation scores in our annotation of deletions, revealing that 2177 hDels remove bases under purifying selection at levels comparable to exonic sequence. As expected from human genetics studies, we found that deletions are depleted for removing conserved sequence ($p < 10^{-3}$), but that in aggregate hDels remove a 116,828 bp of sequence exceeding exon-levels of conservation, highlighting possible functional implications of the class. However, we find no evidence of specific categories of genes being disproportionately affected by *cis*-regulatory sequences within hDels when controlling for background sets.

We performed Gene Ontology (GO) term enrichment testing on genes located near hDels (within 1kb, 5kb, 10kb, or 100kb) using all genes as background and found no significant results.

We also tested for enrichment of gene categories near candidate functional elements within hDels, versus candidate functional elements in general. Using iPS cell Tn5-accessible regions as a proxy for functional sequence, we detected no enriched GO terms on genes located near Tn5-accessible regions within hDels versus near all Tn5-accessible regions. Seeking to further narrow the list of candidate functional sequences, we repeated our enrichment testing using runs of conserved bases (at least 8 contiguous bases with phyloP score ≥ 0.95 from a 240 mammalian genome alignment) within Tn5-accessible regions to model conserved elements. There was no enrichment of genes near runs of conserved bases within Tn5-accessible regions within hDels, using runs of conserved bases within all Tn5-accessible regions as background (we again used maximum distances of 100kb, 10kb, 5kb, and 1kb). Thus, although functional elements are typically observed to be enriched near developmental regulatory genes (e.g., conserved non-coding sequences, including HARs, against a genomic background), deleted functional elements were not enriched in specific categories when controlling for the background set of functional elements.

To check whether Tn5-accessible regions might contain binding motifs for specific classes of transcription factors, we located candidate transcription factor binding motifs in hDels (using FIMO with the JASPAR 2020 database). We then tested for enrichment GO terms on transcription factors with predicted binding motifs in hDels, using all transcription factors in the JASPAR 2020 database as background, and again found no enriched terms.

To test whether genes found in this experiment to be significantly differently regulated by sequences within hDels were enriched for any functional categories, we tested genes significantly perturbed by hDel CRISPRi for GO enrichment, using both all perturbed genes and all genes within 100 kb of hDels as background. We again found no significant results. This is likely due in part to the small number of genes that passed our significance threshold. But we do note, however, that *PLPP1* and *CERK* are in the same lipid metabolism pathway.

2. Further, since we have a set of functional hDels, are there any general rules that can be extracted from their sequence/genomic features? E.g., hConDels vs. non-conserved. Or proximity to genes. Currently, the study narrows in on a few interesting examples, but ideally, generalizable rules of hDels would be useful to the community.

We agree that a set of general rules would be useful for the community, but we cannot identify sequence features that make functional hDels (of the type that were linked to genes using Perturb-seq) readily identifiable from other, non-functional hDels. Functional hDels are similar distances to the nearest gene as other hDels (K-S p -value 0.52 and 0.13 for hDel-v3 and hDel-v4 library hits, respectively). Furthermore, as stated above, there are no enriched functional categories of genes near hDels with regulatory potential when controlling for the background set of elements with regulatory potential.

We do note that 3 of 20 hDels found to contain at least one run of conserved bases (at least 8 contiguous bp with phyloP score ≥ 0.95 from a 240 mammalian genome alignment) contained significant CRISPRi guide target sequence (of 26 total significant hDels). This is a significant

enrichment ($p \sim 6 \times 10^{-7}$) and suggests that evolutionary conservation could be an indicator of regulatory potential in other hDels.

3. Do authors have details on the population prevalence of identified hDels, particularly the ones they suggest are functional? In Kronenberg, et al they narrow in on ~5000 “fixed” fhDels, but here the study focuses on >7000. It’s not clear where the additional 2000 comes from (polymorphic?). Are fixed hDels more likely to show a function or have defining genomic features?

Reviewer 1 is correct in highlighting the different sets of hDels reported with evidence of fixation in Kronenberg et al. The Kronenberg et al. main text focused on 5,892 fhDels. The fixed determination involved comparing two new fully assembled human genomes to the reference genome and genotyping of SVs in at least two individuals from each SGDP continental group ($N = 16$). However, the list of 5,892 hDels further required the deletion to also be a unique structural variant at the locus among apes (11 orangutans, 8 gorillas, 8 chimpanzees, and 2 bonobos). This criterion eliminated fixed deletions in humans that had undergone independent events in other apes. For example, a human-specific hCONDEL near androgen receptor, that removes a penile spine enhancer and previously validated as fixed among humans (McLean et al., Nature 2011) did not pass the multispecies alignment (MSA) determinations as fixed because there is an independent and distinct structural variant involving an inversion in gorilla. However, this hCONDEL still represents a derived and fixed human-specific deletion.

To include human-specific deletions that may have independent events in other ape lineages, the Kronenberg paper also includes 1,508 additional fhDels in supplementary table 11.1. We elected to use this broader set as McLean et al. had already shown an intriguing functional example and we aimed for a more comprehensive survey of a class of human-specific variants. We further merged results with 583 hCONDELs, including 76 with evidence for polymorphism (13%) from McLean et al. to take the union of deletions from these two studies, and we compared all deletions with the UCSC hg38-panTro6 net alignment, resolving any differences by reciprocal BLAT to generate the list of 7,282 for further analysis. After merging hDels and hCONDELs and consolidating deletions separated by <1 kb, our dataset included 1504 hDels that were not previously called as fixed by MSA in Kronenberg (with the requirement for fixed copy number in other apes), and 5774 that were. Identification of functional hDels did not significantly overlap with the categorization of hDels as MSA confirmed. Of 20 hDels linked to target genes, 13 were previously determined as fixed by MSA, which is not a significant enrichment compared to MSA confirmed fixed deletions in the overall set by Fisher’s exact test ($p = 0.16$). Of these 20 hDels, there is evidence that one, hDel_76, is polymorphic, which is listed in Supplemental table 1.

4. In the hDel-v3 Perturb-seq, how do authors account for the possible biases introduced by pseudo-bulking across cells when testing functional impacts of hDels impacting proliferation? Might technical artefacts arise from genes with fewer cells (reduced proliferation) vs. more cells (increased proliferation), resulting in systematically fewer transcripts per gene after summing.

We have used a pseudobulk approach for differential expression testing because methods that compare individual cells frequently have inflated type 1 error rates (Squiar et al., Nature Communications 2021; Zimmerman et al., Nature Communications 2021). Reviewer 1 correctly points out that the number of cells per sgRNA varies in Perturb-seq and presents a challenge for differential expression testing. To account for a variable number of cells in each +sgRNA pseudobulk, DESeq2 uses size factors to make comparisons of gene expression between groups (in this case +hDel-targeting sgRNA vs. +non-targeting sgRNA).

Minor comment:

1. For hDel-v3 Perturb-seq, only DE genes within 100 kbp of hDel are considered a target. How many hDel/gene combinations fit this criterion in the assay and was this factored into the model when determining statistical significance?

For hDel-v3, we identified 4 hDel-gene pairs within 100 kb (FDR < 0.1), including hDel_6012-*RPL26*, hDel_349-*MRPS14*, and hDel_6304-*MBD3*. For multiple hypothesis correction, the Benjamini-Hochberg procedure was applied to all sgRNA-gene pairs separated by ≤ 100 kb. The approach for assessing statistical significance is outlined below (inspired by Cooper et al., Science 2022):

- sgRNA-gene pairs were Z-score normalized using the DESeq2 \log_2 fold-change divided by the DESeq2 standard error of the \log_2 fold-change
- p -values were calculated from the survival function of a normal distribution for each gene within 100 kb of any hDel, drawing a null distribution from all sgRNA-gene pairs separated by ≥ 100 kb
- the Benjamini-Hochberg procedure was applied to all sgRNA-gene pairs separated by ≤ 100 kb

Reviewer #2 (Remarks to the Author):

Fair et al developed multiple CRISPRi screening libraries to investigate functions of human-specific deletions during evolution. Using both bulk and single-cell CRISPRi screenings, multiple human-specific deletions either affect cell proliferation in chimpanzee stem cells, or regulate cell gene expression. Studying these regions may reveal how non-coding regions play a role during evolution. In general, the idea is interesting and the approaches are sound. However, the research lacks key validation experiments to evaluate the overall quality of the screening and related conclusions. The authors should also proofread their manuscript, as several main figures were mislabeled.

Major

1. Related to sgRNA and hDel enrichment calculations in Figure 1, it is unclear what is the relationship between DESeq2 analysis on sgRNA and RRA analysis on hDel region enrichment.

Would both data were used in later analysis? If a 250-b step size is used, some sgRNAs may be counted multiple times, would this affect the analysis?

sgRNA-level analysis was performed using DESeq2. To aggregate information from adjacent hDel-targeting sgRNAs, sgRNA-level adjusted p-values from DESeq2 were used as input for α -RRA analysis. This approach identifies significant genomic windows by considering multiple sgRNAs and is conceptually similar to MAGeCK, but uses DESeq2 for negative-binomial regression. Yes, as suggested by Reviewer 2, some sgRNAs are included in adjacent overlapping 500-bp windows; p-values corresponding to all 500-bp windows are accounted for in multiple hypothesis correction by α -RRA.

2. It is unclear about the differences between V1 and V2 tiling. The author need to better describe it. Is the V2 library targeting the hits from V1?

We thank Reviewer 2 for requesting this clarification. Yes, hDel-v2 targets hDels that were identified using hDel-v1. Additionally, hDel-v2 features reduced spacing between proximal sgRNAs (median 7 bp between sgRNAs) compared to hDel-v1 (median 52 bp between sgRNAs) and a greater number of sgRNAs per hDel (hDel-v2, median 119 sgRNAs per hDel; hDel-v1, median 14 sgRNAs per hDel).

We have revised the main text as follows to clarify the transition between hDel-v1 and hDel-v2:

To fine-map functional sequence within proliferation-modifying hDels identified using hDel-v1, we designed a second tiling library (hDel-v2) with reduced spacing between proximal sgRNAs. Because reduced spacing supports the discovery of hDels that may not have been detected using hDel-v1, we included hundreds of hDels with a single proliferation-modifying sgRNA. We omitted a genomic window strategy for sgRNA selection to maximize tiling density, resulting in a library of 78,270 sgRNAs targeting 558 hDels (Fig. 2a, median 7 bp between sgRNAs, median 119 sgRNAs per hDel). As with hDel-v1, we transduced chimpanzee CRISPRi iPS cells (C3624K) with the lentiviral hDel-v2 sgRNA library, selected and cultured sgRNA-expressing cells, and quantified sgRNA enrichment and depletion by high-throughput sequencing.

3. Related to Figure 2c, how is it possible to have 18,148 sgRNAs in the scatter plot if the V2 library only contains 78,270 sgRNAs in total? What are compared?

Figure 2c shows the correlation in fold-change for the 18,148 sgRNAs that were independently screened in hDel-v1 and in hDel-v2. The remaining targeting sgRNAs in hDel-v2 were not screened in hDel-v1, reflecting the greater tiling density in hDel-v2, and cannot be shown on this scatterplot.

4. If G4 and C4 would introduce strong off-target effects, why did not exclude these sgRNAs from V1 analysis? Furthermore, experimental evidence would be needed to show these sgRNAs introduce strong off-target effects, for instance, using discovery seq or other methods.

The large number of sgRNAs containing G₄/C₄ homopolymers in hDel-v2 (Figure S7; $n = 275$ to 404 sgRNAs with G₄ or C₄ homopolymers per spacer position) allowed us to discover position-dependent off-target effects that have been overlooked by previous studies. We organized our presentation of the data based on the order of the experiments, as the design of the hDel-v2 library was based on hDel-v1 as analyzed using state-of-the-art software. We believe that incorporating findings of novel off-target effects discovered using hDel-v2 into analysis of hDel-v1 would have been confusing for the reader, but we ensured that these sgRNAs were removed from hDel-v2 analysis that is presented as validation in Figure 2, controlling for the observed off-target toxicity. We show in Figure S7 that G₄-associated toxicity is also present in a widely-cited CRISPRi tiling study across *GATA1* and *MYC* in human K562 cells (Fulco et al., Science 2016). Although characterizing the source of sgRNA-mediated off-target effects is not the focus of our study, we agree that it is an important direction for future studies. We believe that sharing the observed sgRNA homopolymer-associated toxicity is of value to the research community.

5. For the V4 analysis, there are multiple different sgRNAs per cell. The authors need to clarify how they correct the effects of these different sgRNAs in individual cells during single-cell RNA-seq analysis.

We thank Reviewer 2 for requesting this clarification. For hDel-v4, we transduced cells at a high multiplicity of infection (median 7 sgRNA per cell, median 495 UMIs per sgRNA per cell), consistent with previous single-cell CRISPRi studies (Gasperini et al., Cell 2019; Xie et al., Cell Reports 2019; Morris et al., Science 2023). We knew based on hDel-v1 and hDel-v2 screens that these hDel-targeting sgRNAs were nonessential, motivating us to conduct a high multiplicity of infection screen to increase power to detect differential gene expression and reduce the number of cells needed for scRNA-seq (Gasperini et al., Cell 2019). Because each cell receives a random combination of sgRNAs, no correction was performed for individual sgRNAs. For differential expression testing, we used a pseudobulk approach because methods that compare individual cells frequently have inflated type 1 error rates (Squiar et al., Nature Communications 2021; Zimmerman et al., Nature Communications 2021).

As an orthogonal analysis approach, we used SCEPTRE (Barry et al., Genome Biology 2021), software designed for high multiplicity-of-infection single-cell CRISPR screens, to test for hDel-gene associations with hDel-v4. We performed a left-sided test using SCEPTRE to detect a reduction in gene expression in cells harboring hDel-targeting sgRNAs. We found strong concordance between the results of our DESeq2 pseudobulk approach and SCEPTRE: 10 of the 11 hDel-gene pairs with reduced expression using our approach were significant using SCEPTRE (FDR < 0.1). SCEPTRE results:

	gene_id	gRNA_id	pair_type	p_value	z_value	log_fold_change	p_val_adj
1567	AGO2	hDel_3813_1_H3K4me1_H3K27ac_CUT_Tag_3	candidate	3.362951e-04	-3.453024	-0.164502	8.561112e-02
5221	ATRX	hDel_7051_1_Omni_ATAC_5	candidate	1.273711e-04	-3.495305	-0.152020	4.005448e-02
5216	ATRX	hDel_7051_1_Omni_ATAC_3	candidate	3.752374e-05	-3.719239	-0.145887	1.543092e-02
5081	CERK	hDel_6842_1_Omni_ATAC_2	candidate	2.401378e-05	-3.600870	-0.303503	1.283777e-02
5091	CERK	hDel_6842_1_Omni_ATAC_3	candidate	1.448654e-06	-4.239422	-0.307548	1.106358e-03
1550	FAM49B	hDel_3779_1_Omni_ATAC_3	candidate	2.220446e-16	-9.261927	-0.244949	1.187050e-12
1554	FAM49B	hDel_3779_1_Omni_ATAC_5	candidate	1.093936e-04	-3.558988	-0.134064	3.898788e-02
498	GIN1	hDel_2358_1_H3K4me1_H3K27ac_CUT_Tag_4	candidate	2.185022e-04	-3.573693	-0.412834	6.147963e-02
2919	GRTP1	hDel_5288_1_Omni_ATAC_3	candidate	2.031989e-05	-4.033069	-0.253735	1.207001e-02
3648	HADHA	hDel_585_1_Omni_ATAC_3	candidate	4.252966e-08	-5.426741	-0.272262	5.684089e-05
3655	HADHA	hDel_585_1_Omni_ATAC_4	candidate	2.124798e-08	-5.560398	-0.226500	3.786390e-05
2824	PCCA	hDel_5262_1_Omni_ATAC_2	candidate	3.356717e-05	-4.168035	-0.268763	1.543092e-02
417	PLPP1	hDel_2247_1_Omni_ATAC_5	candidate	2.834527e-04	-3.266120	-0.269346	7.576690e-02
423	PLPP1	hDel_2247_2_Omni_ATAC_1	candidate	3.685669e-07	-4.389916	-0.381037	3.940717e-04
429	PLPP1	hDel_2247_2_Omni_ATAC_2	candidate	3.704883e-05	-3.935323	-0.329190	1.543092e-02
453	PLPP1	hDel_2247_3_Omni_ATAC_2	candidate	2.972447e-06	-3.912249	-0.357502	1.986338e-03
393	PLPP1	hDel_2247_1_Omni_ATAC_1	candidate	2.112338e-04	-3.099292	-0.394377	6.147963e-02
411	PLPP1	hDel_2247_1_Omni_ATAC_4	candidate	1.231457e-04	-3.710434	-0.351117	4.005448e-02
220	ST6GAL1	hDel_1572_1_H3K4me1_H3K27ac_CUT_Tag_3	candidate	1.032569e-04	-3.788877	-0.225646	3.898788e-02
210	ST6GAL1	hDel_1572_1_H3K4me1_H3K27ac_CUT_Tag_1	candidate	6.902294e-07	-5.168507	-0.397954	6.149944e-04
84	SUCLG2	hDel_1273_1_H3K4me1_H3K27ac_CUT_Tag_1	candidate	1.033357e-11	-6.553732	-0.317530	2.762163e-08
516	ZNF300	hDel_2487_1_Omni_ATAC_1	candidate	3.831416e-04	-3.225716	-0.564989	9.310340e-02

6. For the data related to hDel_2247, why mouse tissues were used for the comparison? What would be the results if human tissues were used?

We thank the reviewer for raising this point as further analysis has led to additional information regarding the function of hDel_2247. Human-specific deletion 2247 (hDel_2247) is not present in the human genome. We originally analyzed epigenomic data from the orthologous sequence in mice because there is no comparable ENCODE dataset in nonhuman primates.

We agree with Reviewer 2 that analyzing sequences closer to the human-chimpanzee common ancestor would be ideal for characterizing the regulatory activity of the deleted sequence. Therefore, we used an orthogonal approach - transient transgenic *lacZ* reporter assays in mouse embryos - to analyze the sufficiency of the chimpanzee sequence for driving reporter gene expression *in vivo* during embryonic mouse development.

We found that the chimpanzee sequence (1,557 bp; chr5:45194159-45195715 in panTro6), distinct from epigenomic signatures in mouse, regulates gene expression in the olfactory bulb and anterior neocortex, two structures that have been modified recently in human brain evolution. We thank Reviewer 2 for highlighting the importance of focusing on sequences closer

to the ancestral allele lost in recent human evolution, as this led to updated findings regarding the anatomical specificity of hDel_2247.

7. There are no individual cell clones made to validate the screening results related to different phenotypes. As genetics screenings would suffer from false positive hits, these validations are key to show the screenings indeed identified proper hits and deletion regions.

We agree that orthogonal approaches provide additional value. However, we are cautious about experiments involving the generation of isogenic cell lines due to the risk of acquiring unintended genetic changes (Baker et al., Nature Biotechnology 2007; Merkle et al., Nature 2017) and phenotypic variability among individual clones (Westermann et al., Scientific Reports 2022). Instead, we used an orthogonal approach, transient transgenic *lacZ* reporter assays in mouse embryos, finding that sequences within hDel_2247 drive reporter gene expression specifically in olfactory bulb the anterior neocortex. As hDel_2247's target gene *PLPP1* hydrolyzes LPA, which is a driver of both olfactory ensheathing cell migration and neural progenitor proliferation (Medelnik et al., Stem Cell Reports 2018), this finding may motivate future examination in the context of human brain evolution.

8. The research aims to study genomic deletions in evolution, the authors would need to use CRISPR editing to delete at least some of the top hits to show the functions of the regions they identified and associated with different genes or phenotypes. Furthermore, non-coding regulatory regions, i.e. enhancer or silencer, may have different functions. CRISPRa may be needed to identify certain types of non-coding deletions, especially loss of silencer activity.

We agree that CRISPRi repression of *cis*-regulatory sequence may not enable detection of all functions of noncoding sequences, and we have now added a sentence to the discussion highlighting this limitation: "For *cis*-regulatory elements with repressive effects on transcription, such as silencers, CRISPR activation-based approaches may be useful for target gene identification." In addition, we now highlight an example of an hDel with apparent silencer activity. Targeting of hDel_1608, which intersects multiple epigenetic features results in increased expression of *C4orf48* (Fig. 4i,j).

9. As the authors also agreed that the CRISPRi system has certain effector size limitations, how do the authors consider such an effect in their design of the smaller scale libraries, such as V2, V3, and V4?

Consistent with previous studies (Fulco et al., Nature Genetics 2019), we find that CRISPRi has a narrow window of activity at promoter-distal sites. We consider this limitation during the design of hDel-v2, hDel-v3, and hDel-v4 by including at least 5 sgRNAs per targeted hDel. For example, 8 sgRNAs targeting hDel_2247 reduce the expression of *PLPP1* in hDel-v4 (Fig. 5a,b).

10. The authors concluded that most hDels are dispensable for proliferation. Would this be a strong statement, as the screenings were only done in one cell type?

We agree that this statement is overly broad. We have revised the sentence as follows:

Together, hDel-v1 and hDel-v2 identify cellular phenotypes for select hDels and indicate that despite the predicted phenotypic importance of SV-sized noncoding deletions, hDels as a class of human-specific SVs are largely dispensable for iPS cell proliferation.

I am not sure if such mistakes are just minor issues, but a few figures were not labeled properly.

Figure 1 b and c were not referred to or referred wrongly.

Figure 2 panels were wrongly labeled.

We thank Reviewer 2 for raising this concern. We have corrected the panel labels and we apologize for any inconvenience this may have caused.

Reviewer #3 (Remarks to the Author):

The manuscript by Fair et al., delves into a pivotal yet underexplored domain within genomics, specifically focusing on human-specific deletions (hDels) and their consequential roles in cellular processes. The study's innovative approach, utilizing cell proliferation and transcriptome changes as a functional readout, provides a fresh perspective in understanding these genomic elements. The authors' application of high throughput CRISPRi screening, executed both at bulk and single-cell levels, stands out as a methodologically robust technique to characterize the role of hDels. Through this screening, the identification of several hDels that influence cell proliferation or gene expression is a significant contribution to the field. Notably, the screening assays have been meticulously designed and executed, demonstrating a high level of scientific rigor. The data are presented with clarity and precision, making the study's findings accessible and comprehensible. The potential implications of these findings in the broader context of genomic research are substantial, indicating that this study, upon addressing specific concerns outlined below, merits timely publication in a prestigious academic journal.

Major Concerns:

1. Validation of hDels Function: The identified hDels from CRISPRi screen require further validation. It is recommended that region-specific deletions be executed using Cas9. This approach would solidify the functional attributions made to these hDels. Alternatively, this concern can also be addressed by analyzing the gene regulatory effects of cis-regulatory hDels in chimpanzee autotetraploids and allotetraploid cells by comparing gene expression between chimpanzee vs human alleles. Such comparative analysis would enhance our understanding of hDels' regulatory function/mechanisms when the sequences are deleted across species rather than epigenetically silenced.

We agree on the importance of orthogonal assays. As suggested by Reviewer 3, we quantified *cis*-regulatory divergence between human and chimpanzee alleles in human-chimpanzee allotetraploid iPS cells for hDel target genes. We found that the human *CERK*, *FAM49B*, *GRT1*, *SUCLG2*, *MRPS14*, and *MDB3* human alleles drove reduced expression compared to chimpanzee alleles in allotetraploid iPS cells (Extended Data Fig. 6h-j, Fig. 4i), consistent with the human-specific loss of *cis*-regulatory sequence and with our results using CRISPRi. As further validation, we used transient transgenic *lacZ* reporter assays in mouse embryos, finding that a 1.5 kb sequence within hDel_2247 drives reporter gene expression specifically in the olfactory bulb and anterior neocortex.

CRISPR-based experiments are complemented by epigenetic characterization of hDel chromatin using Omni ATAC-seq and H3K4me1, H3K4me3, H3K27ac, and H3K27me3 CUT&Tag.

2. For hDels identified from Perturb-seq and acting as candidate *cis*-regulatory elements (cCREs), what are the linear distance between these hDels and their target genes? Do these hDels function as distal cCREs? If so, does chromatin looping data in iPSC or hESC support such distal regulation? Given the availability of scRNA-seq data, analyses focusing on both local and distal regulation would deepen our understanding of hDel functions.

We thank Reviewer 3 for requesting this clarification. The linear distance between hDel-targeting sgRNA and TSS for corresponding *cis* target gene is plotted on a log₁₀ scale in Fig. 4d. Yes, these hDels function as intergenic or intronic *cis*-regulatory elements; we provide a table of linear distances between hDel-targeting sgRNAs and their target genes for clarification:

	gene_name	baseMean	log2FoldChange	lfcSE	sg_ID	hDel_ID	BH_pvalue	sgRNA_chr	sgRNA_start	sgRNA_end	gene_chr	TSS_start	TSS_end	minimum_distance_TSS
17	CERK	237.041216	-0.390854	0.181399	hDel_6842_1_Omni_ATAC_5	hDel_6842	8.475397e-02	chr22	29570055	29570076	chr22	29515218	29571032	956
19	CERK	339.174828	-0.439208	0.129208	hDel_6842_1_Omni_ATAC_3	hDel_6842	4.112361e-04	chr22	29569658	29569679	chr22	29515218	29571032	1353
18	CERK	300.063508	-0.468266	0.148447	hDel_6842_1_Omni_ATAC_2	hDel_6842	1.601036e-03	chr22	29569418	29569439	chr22	29515218	29571032	1593
14	C4orf48	1256.319698	0.260029	0.073670	hDel_1608_1_Omni_ATAC_1	hDel_1608	2.557768e-02	chr4	1881391	1881412	chr4	1877750	1879726	1665
13	C4orf48	1503.846138	0.217073	0.073532	hDel_1608_1_Omni_ATAC_5	hDel_1608	9.891949e-02	chr4	1881635	1881656	chr4	1877750	1879726	1909
26	DRAP1	1948.107106	0.151447	0.056553	hDel_4544_1_Omni_ATAC_4	hDel_4544	7.644465e-02	chr11	61887971	61887992	chr11	61882532	61884808	3163
21	LINC01355	63.754345	0.805228	0.345491	hDel_76_1_Omni_ATAC_3	hDel_76	9.891949e-02	chr1	21848838	21848859	chr1	21856555	21861825	7696
9	AC011447.7	1006.331154	0.284872	0.079125	hDel_6391_1_Omni_ATAC_4	hDel_6391	9.891949e-02	chr19	20738568	20738589	chr19	20754478	20756070	15889
29	AGO2	1167.784582	-0.211801	0.080623	hDel_3813_1_H3K4me1_H3K27ac_CUT_Tag_3	hDel_3813	2.621160e-03	chr8	138674167	138674188	chr8	138579544	138695042	20854
25	HADHA	1175.448713	-0.334633	0.074846	hDel_585_1_Omni_ATAC_4	hDel_585	1.570621e-05	chr2p	26190636	26190657	chr2A	26158517	26212765	22108
24	HADHA	921.869241	-0.411723	0.100714	hDel_585_1_Omni_ATAC_3	hDel_585	1.422987e-04	chr2p	26190623	26190644	chr2A	26158517	26212765	22121
15	SUCLG2	827.598430	-0.516090	0.088187	hDel_1273_1_H3K4me1_H3K27ac_CUT_Tag_1	hDel_1273	9.806344e-10	chr3	67828366	67828387	chr3	67553015	67852971	24584
11	ST6GAL1	447.502223	-0.528293	0.129451	hDel_1572_1_H3K4me1_H3K27ac_CUT_Tag_1	hDel_1572	2.837788e-05	chr3	185043277	185043298	chr3	185012959	185161534	30318
12	ST6GAL1	581.106116	-0.266117	0.110265	hDel_1572_1_H3K4me1_H3K27ac_CUT_Tag_3	hDel_1572	4.762668e-02	chr3	185043887	185043908	chr3	185012959	185161534	30928
10	ST6GAL1	444.500490	-0.334289	0.158225	hDel_1572_1_H3K4me1_H3K27ac_CUT_Tag_5	hDel_1572	9.891949e-02	chr3	185044154	185044175	chr3	185012959	185161534	31195
5	PLPP1	231.887286	-0.489614	0.190596	hDel_2247_1_Omni_ATAC_1	hDel_2247	2.864017e-02	chr5	45194281	45194302	chr5	45013052	45123031	71250
3	PLPP1	283.017463	-0.409713	0.155129	hDel_2247_1_Omni_ATAC_4	hDel_2247	2.557768e-02	chr5	45194434	45194455	chr5	45013052	45123031	71403
8	PLPP1	319.150735	-0.324816	0.145496	hDel_2247_1_Omni_ATAC_5	hDel_2247	7.644465e-02	chr5	45194470	45194491	chr5	45013052	45123031	71439
4	PLPP1	312.843675	-0.432282	0.143457	hDel_2247_2_Omni_ATAC_1	hDel_2247	5.414855e-03	chr5	45194789	45194810	chr5	45013052	45123031	71758
2	PLPP1	327.857390	-0.351273	0.134106	hDel_2247_2_Omni_ATAC_2	hDel_2247	2.557768e-02	chr5	45194997	45195018	chr5	45013052	45123031	71966
7	PLPP1	367.825819	-0.284491	0.120956	hDel_2247_2_Omni_ATAC_3	hDel_2247	5.640323e-02	chr5	45195035	45195056	chr5	45013052	45123031	72004
1	PLPP1	306.823221	-0.350859	0.150508	hDel_2247_2_Omni_ATAC_4	hDel_2247	5.640323e-02	chr5	45195499	45195520	chr5	45013052	45123031	72468
6	PLPP1	286.957492	-0.464068	0.153315	hDel_2247_3_Omni_ATAC_2	hDel_2247	5.414855e-03	chr5	45196682	45196703	chr5	45013052	45123031	73651
20	PEX13	225.751548	0.373572	0.178488	hDel_698_1_Omni_ATAC_2	hDel_698	9.891949e-02	chr2p	61082183	61082204	chr2A	61161444	61196558	79240
28	FAM49B	1527.362071	-0.220769	0.069554	hDel_3779_1_Omni_ATAC_5	hDel_3779	4.092563e-02	chr8	127962912	127962933	chr8	127867587	128046465	83532
27	FAM49B	2139.422427	-0.388141	0.059877	hDel_3779_1_Omni_ATAC_3	hDel_3779	5.540823e-11	chr8	127962895	127962916	chr8	127867587	128046465	83549
23	GRT1	688.901489	-0.302068	0.107908	hDel_5288_1_Omni_ATAC_3	hDel_5288	5.640323e-02	chr13	94730930	94730951	chr13	94602190	94641235	89695
0	FAT3	399.540133	-0.389199	0.136377	hDel_4623_1_H3K4me1_H3K27ac_CUT_Tag_5	hDel_4623	9.891949e-02	chr11	88224148	88224169	chr11	88120462	88793382	103686
16	ATRX	2205.672124	-0.234513	0.060973	hDel_7051_1_Omni_ATAC_3	hDel_7051	7.907832e-02	chrX	73154246	73154267	chrX	73035475	73320279	118771
22	PCCA	464.533625	-0.387194	0.109791	hDel_5262_1_Omni_ATAC_2	hDel_5262	2.897691e-03	chr13	81577960	81577981	chr13	81318141	81764169	186188

For differential expression testing, we first determine whether any hDel alters the expression of any expressed gene within a linear distance of ≤ 100 kb (to gene body) to establish a *cis*-regulatory linkage. For hDels with identified *cis*-regulatory linkages, we then performed a genome-wide *trans* differential expression test. For example, we identified 83 differentially expressed genes upon hDel_6304 CRISPRi, including transcription factors controlling meso-endoderm differentiation (*EOMES*, *GATA6*, *LHX1*, *MIXL1*), all of which were down-regulated (Fig. 3k, FDR < 0.1).

We have also intersected hDels linked to target genes using Perturb-seq with chimpanzee iPS cell chromatin conformation (Hi-C) from Eres et al., PLoS Genetics 2019. We find that all 20 hDels linked to target genes using Perturb-seq are within regions of 3D genomic interactions in chimpanzee iPS cells. Half of these hDels (10 hDels) are within regions that contact the TSS of the linked gene.

3. The 6.8 kb intergenic deletion hDel_2247 necessitates further elucidation. Since the typical effect size per gRNA in CRISPRi is usually less than 1kb, additional details characterizing the influence of sgRNAs on this larger region are needed. Are there specific motifs or subregions within this 6.8kb area that are critical for its function?

We have performed transient transgenic *lacZ* reporter assays in mouse embryos to characterize the *cis*-regulatory capacity of sequences within hDel_2247 *in vivo*. We find that sequences within hDel_2247 intersecting Omni ATAC-seq, H3K4me1, and H3K27ac drive reporter gene expression specifically in the olfactory bulb and anterior neocortex in multiple embryos ($n = 4$ of 5 transgenic embryos). These results identify a subregion within hDel_2247 that is critical for its *cis*-regulatory function (1,557 bp; chr5:45194159-45195715 in panTro6) and provide independent validation for the 8 sgRNAs targeting hDel_2247 within this sequence that reduce the expression of *PLPP1* in hDel-v4 (Fig. 4f).

4. The text within several figures is too small to be easily readable.

We thank Reviewer 3 for raising this concern and we apologize for any inconvenience this may have caused. We have adjusted font size throughout the main figures.

5. There appears to be a discrepancy between the data presented in the figures and the conclusions drawn in the text. For instance, supporting data for the findings discussed in sections such as “We next examined RPL26 MRPS14, and MBD3...” and “sgRNAs targeting a putative inhibitor of DNA binding ID1 *cis*-regulatory element reduced the expression of ID1...” are not evident. Please include and clearly cite relevant figures or supplementary figures to substantiate these conclusions.

We thank Reviewer 3 for raising this concern and we apologize for omitting these figures. We have added supporting data for *MRPS14*, *MBD3*, and *RPL26* in Extended Data Fig. 6h-j. We have also added supporting data for *ID1* in Extended Data Fig. 8d.

RESPONSE TO REVIEWERS' COMMENTS

Reviewer #2 (Remarks to the Author):

It is very common to validate some of the top hits from most of the genetic screenings to evaluate the performance of the screening, as both referees 2 and 3 pointed out. Especially that the authors would like to use CRISPRi to study the effects of genomic deletions.

We appreciate this reviewer's helpful comments.

We have used pairs of Cas9 RNPs to generate genomic deletions of two hDels (hDel_2247 and hDel_3779) that are linked to gene targets in our hDel-v4 CRISPRi Perturb-seq screen. We find that deletion of hDel_2247 reduces the expression of *PLPP1* 65.0% to 75.9% in three independent polyclonal populations of chimpanzee iPS cells. Similarly, we find that deletion of hDel_3779 reduces the expression of *FAM49B* 9.1% to 27.6%.

Related to that, do the 16 hDels identified from V1 screening overlap with the 38 hDels from V2?

We assigned hDel-targeting sgRNAs to overlapping 500-bp genomic windows (250-bp step size) and combined sgRNAs into hDel FDRs using alpha-robust rank aggregation (α -RRA) from MAGeCK to identify proliferation-modifying hDels. 4 of 16 hDels from hDel-v1 overlap with hDel-v2. This discrepancy is not a matter of data quality: hDel-targeting sgRNAs screened in both hDel-v1 and hDel-v2 were highly correlated ($r = 0.73$, Fig. 2c). Instead, the

addition of new sgRNAs in hDel-v2 did not show similar proliferation-modify effects, partly due to the fact that hDel-v1 sgRNAs were selected for high on-target activity (using DeepHF), whereas hDel-v2 sgRNAs were only filtered for off-target specificity (no on-target activity preference). All major conclusions we draw from the results of our study further depend upon hDel-gene linkages from Perturb-seq (hDel-v3 and hDel-v4).

It is also puzzling to see CRISPRi (with repressive activity) targeting a silencer (which probably regulated by a repressive TF) will lead to transcriptional activation, as showed by dDel_1608 and C4orf48.

hDel_1608 intersects a Tn5-accessible region (panel i) and using conservative pseudobulk and two-tailed statistical tests for analysis of Perturb-seq (hDel-v4), 2 sgRNAs increase the expression of C4orf48 (panel j, 16.2 to 19.8%, FDR < 0.1). This finding is consistent with the negative regulation of mRNA by a divergently transcribed lncRNA (<https://pubmed.ncbi.nlm.nih.gov/32777522/>). We now acknowledge that other scenarios are also possible, including CRISPRi preventing the binding of repressive transcription factors to target DNA.